# ADIR: Adaptive Diffusion for Image Reconstruction

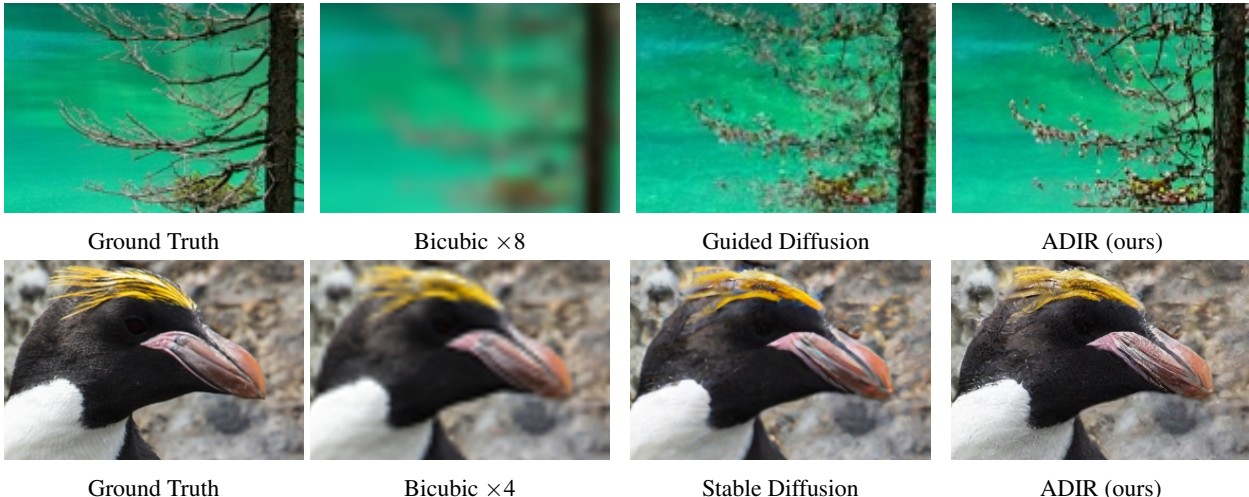

|  |  |  |  |
|---|---|---|---|
| Ground Truth | Bicubic ×8 | Guided Diffusion | ADIR (ours) |
| Ground Truth | Bicubic ×4 | Stable Diffusion | ADIR (ours) |

Figure 1: Super-resolution with scale factors 4 and 8, using Stable Diffusion (Rombach et al., 2022), Guided Diffusion (Dhariwal & Nichol, 2021), and our method ADIR. The adaptability of ADIR allows reconstructing finer details.

## Abstract

In recent years, denoising diffusion models have demonstrated outstanding image generation performance. The information on natural images captured by these models is useful for many image reconstruction applications, where the task is to restore a clean image from its degraded observation. In this work, we propose a conditional sampling scheme that exploits the prior learned by diffusion models while retaining agreement with the measurements. We then combine it with a novel approach for adapting pre-trained diffusion denoising networks to their input. We perform the adaptation using images that are "nearest neighbours" to the degraded image, retrieved from a diverse dataset using an off-the-shelf visual-language model. To evaluate our method, we test it on two state-of-the-art publicly available diffusion models, Stable Diffusion and Guided Diffusion. We show that our proposed **A**daptive **D**iffusion for **I**mage **R**econstruction (**ADIR**) approach achieves significant improvement in image reconstruction tasks. Our code will be available online upon publication.

## 1 Introduction

Image reconstruction problems appear in a wide range of applications, where one would like to reconstruct an unknown clean image $\mathbf{x} \in \mathbb{R}^n$ from its degraded version $\mathbf{y} \in \mathbb{R}^m$, which can be noisy, blurry, low-resolution, etc. The acquisition (forward) model of $\mathbf{y}$ in many important degradation settings can be formulated using the following linear model

$$\mathbf{y} = \mathbf{A}\mathbf{x} + \mathbf{e}, \tag{1}$$

where $\mathbf{A} \in \mathbb{R}^{m \times n}$ is the measurement operator (blurring, masking, sub-sampling, etc.) and $\mathbf{e} \in \mathbb{R}^m \sim \mathcal{N}(0, \sigma^2 \mathbf{I}_m)$ is the measurement noise. Typically, just fitting the observation model is not sufficient for recovering $\mathbf{x}$ successfully; thus, prior knowledge of the characteristics of $\mathbf{x}$ is needed.

Over the past decade, many works suggested solving the inverse problem in Eq. equation 1 using a single execution of a deep neural network, trained using pairs of clean $\{\mathbf{x}_i\}$ images and their degraded versions $\{\mathbf{y}_i\}$ obtained by applying the forward model equation 1 on $\{\mathbf{x}_i\}$ (Dong et al., 2015; Sun et al., 2015; Lim et al., 2017; Zhang et al., 2017a; Lugmayr et al., 2020; Liang et al., 2021). However, these approaches tend to overfit the observation model and perform poorly on setups that have not been considered in training and several methods have been proposed to overcome that (Shocher et al., 2018; Tirer & Giryes, 2019; Hussein et al., 2020b; Ji et al., 2020; Wei et al., 2020; Wang et al., 2021; Zhang et al., 2021b; 2022). Tackling this limitation with dedicated training for each application is not only computationally inefficient but also often impractical.

Several approaches such as Deep Image Prior (Ulyanov et al., 2018), zero-shot-super-resolution (Shocher et al., 2018) or GSURE-based test-time optimization (Abu-Hussein et al., 2022) rely solely on the observation image $\mathbf{y}$. They utilize the implicit bias of deep neural networks and gradient-based optimizers, as well as the self-recurrence of patterns in natural images when training a neural model directly on the observation and in this way reconstruct the original image. Although these methods are not limited to a family of observation models, they usually perform worse than data-driven methods, since they do not exploit the robust prior information that the unknown image $\mathbf{x}$ share with external data that may contain images of the same kind. The alternative popular approach that exploits external data while remaining flexible to the observation model, uses deep models for imposing only the prior. It typically uses pretrained deep denoisers (Zhang et al., 2017b; Arjomand Bigdeli et al., 2017; Tirer & Giryes, 2018; Zhang et al., 2021a) or generative models (Bora et al., 2017; Dhar et al., 2018; Hussein et al., 2020a) within the optimization scheme, where consistency of the reconstruction with the observation $\mathbf{y}$ is maintained by minimizing a data-fidelity term.

Recently, diffusion models (Dhariwal & Nichol, 2021; Nichol & Dhariwal, 2021; Sohl-Dickstein et al., 2015; Ho et al., 2020) have shown remarkable capabilities in generating high-fidelity images and videos (Ho et al., 2022). These models are based on a Markov chain diffusion process performed on each training sample. They learn the reverse process, namely, the denoising operation between each two points in the chain. Sampling images via pretrained diffusion models is performed by starting from a pure white Gaussian noise image, then progressively denoise and sample a less noisy image, until reaching a clean image. Since diffusion models capture prior knowledge of the data, one may utilize them as deep priors/regularization for inverse problems(Song et al., 2021; Lugmayr et al., 2022; Avrahami et al., 2022b; Kawar et al., 2022a; Choi et al., 2021; Rombach et al., 2022).

In this work, we propose an Adaptive Diffusion framework for Image Reconstruction (ADIR). First, we devise a diffusion guidance sampling scheme that solves equation 1 while restricting the reconstruction of $\mathbf{x}$ to the range of a pretrained diffusion model. Our scheme is based on novel modifications to the guidance used in (Dhariwal & Nichol, 2021) (see Section 3.2 for details). Then, we propose a technique that uses the observations $\mathbf{y}$ to adapt the diffusion network to patterns beneficial for recovering the unknown $\mathbf{x}$. Adapting the model's parameters is based on $K$ external images similar to $\mathbf{y}$ in some neural embedding space that is not sensitive to the degradation of $\mathbf{y}$. These images are retrieved from a diverse dataset and the embedding can be calculated using an off-the-shelf encoder model for images such as CLIP (Radford et al., 2021).

In this work, ADIR is mainly developed for image reconstruction tasks. Yet, we also showcase that the ADIR adaptation strategy can be employed for text-guided image editing. Note that for the latter, we just show the potential of our strategy and that it can be combined with existing editing techniques. We leave further exploration of the use of ADIR to editing to a future work.

The contribution of the ADIR framework is the proposal of an *adaptive* diffusion approach to inverse problems. We evaluate it with two state-of-the-art diffusion models: Stable Diffusion (Rombach et al., 2022) and Guided Diffusion (Dhariwal & Nichol, 2021), and show that it outperforms existing methods in the super-resolution and deblurring tasks.

## 2 Related Work

**Diffusion models** In recent years, many works utilized diffusion models for image manipulation and reconstruction tasks (Choi et al., 2021; Rombach et al., 2022; Kawar et al., 2022b;a; Whang et al., 2022; Saharia et al., 2022b; Zhu et al., 2023; Özdenizci & Legenstein, 2023; Delbracio & Milanfar, 2023; Garber & Tirer, 2023), where a denoising network is trained to learn the prior distribution of the data, then at test time, some conditioning mechanism is combined with the learned prior to solve very challenging imaging tasks (Avrahami et al., 2022b;a; Chung et al., 2022a).

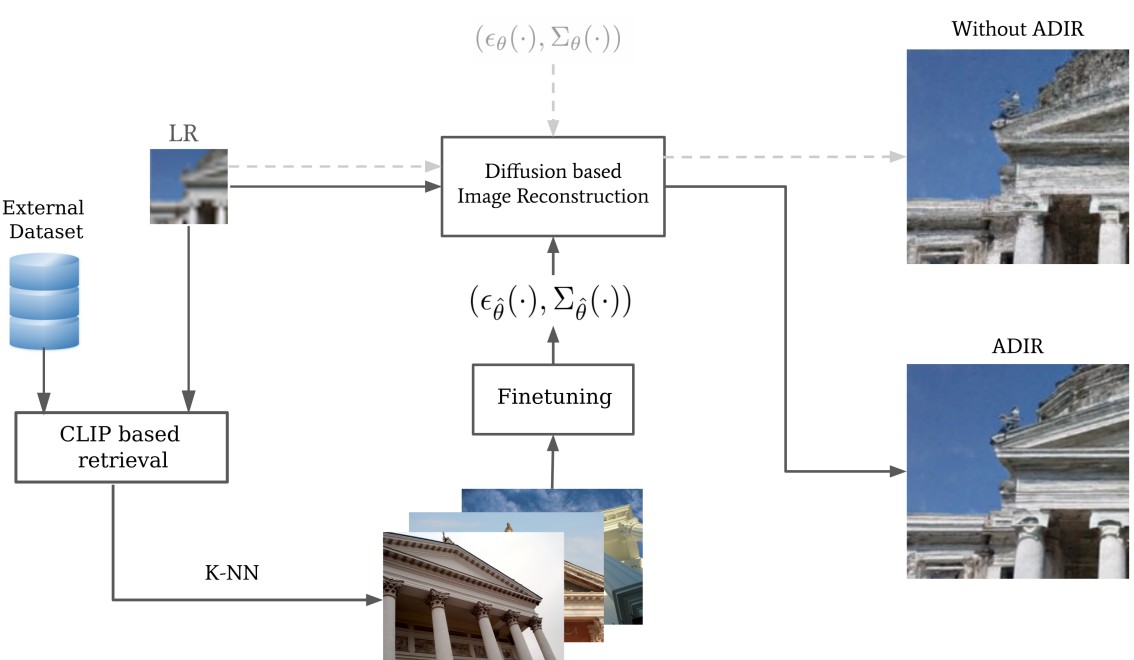

Figure 2: Diagram of our proposed method ADIR (Adaptive Diffusion for Image Reconstruction) applied to the super resolution task. Given a pretrained diffusion model $(\epsilon_\theta(\cdot), \Sigma_\theta(\cdot))$ and a Low Resolution (LR) image, we look for the $K$ nearest neighbor images to the LR image, then using ADIR we adapt the diffusion model and use it for reconstruction.

Note that our novel adaptive diffusion ingredient can be incorporated with any conditional sampling scheme that is based on diffusion models.

In (Whang et al., 2022; Saharia et al., 2022b) the problems of deblurring and super-resolution were considered. Specifically, a diffusion model is trained to perform the task. In this way, the model learns to carry out the deblurring or super-resolution task directly. Notice that these models are trained for one specific task and cannot be used for the other as is.

The closest works to us are (Giannone et al., 2022; Sheynin et al., 2022; Kawar et al., 2022b). These very recent concurrent works consider the task of image editing and perform an adaptation of the used diffusion model using the provided input and external data. Yet, notice that neither of these works consider the task of image reconstruction as we do here or apply our proposed sampling scheme for this task.

**Image-Adaptive Reconstruction** Adaptation of pretrained deep models, which serve as priors in inverse problems, to the unknown true $\mathbf{x}$ through its observations at hand was proposed in (Hussein et al., 2020a; Tirer & Giryes, 2019). These works improve the reconstruction performance by fine-tuning the parameters of pretrained deep denoisers (Tirer & Giryes, 2019) and GANs (Hussein et al., 2020a) via the observed image $\mathbf{y}$ instead of keeping them fixed during inference time. The image-adaptive GAN (IAGAN) approach (Hussein et al., 2020a) has led to many follow up works with different applications, e.g., (Bhadra et al., 2020; Pan et al., 2021; Roich et al., 2022; Nitzan et al., 2022). Recently, it has been shown that one may even fine-tune a masked-autoencoder to the input data at test-time for improving the adaptivity of classification neural networks to new domains (Gandelsman et al., 2022).

In this paper we consider test-time adaptation of diffusion models for inverse problems. As far as we know, adaptation of diffusion models has not been proposed in image reconstruction tasks. Furthermore, while existing works fine-tune the deep priors directly using $\mathbf{y}$, we propose an improved strategy where the tuning is based on $K$ external images similar to $\mathbf{y}$ that are automatically retrieved from an external dataset.

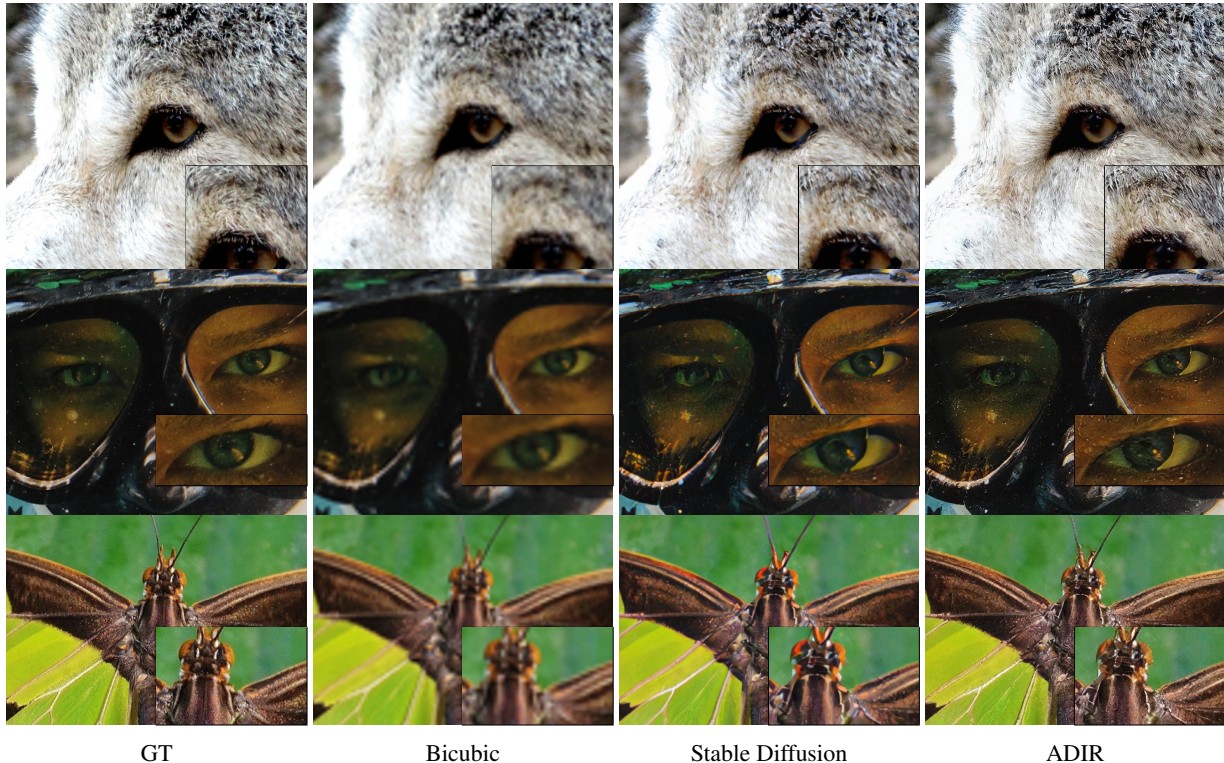

|  |  |  |  |
|:---:|:---:|:---:|:---:|
| GT | Bicubic | Stable Diffusion | ADIR |

Figure 3: Comparison of super resolution ($256^2 \rightarrow 1024^2$) results of Stable Diffusion model(Rombach et al., 2022) and our method (ADIR). As can be seen from the images, our method outperforms Stable Diffusion in both sharpness and reconstructing details.

## 3 Method

We now turn to present our proposed approach. We start with a brief introduction to regular denoising diffusion models. After that we describe our proposed strategy for modifying the sampling scheme of diffusion models for the image reconstruction task. Finally, we present our suggested adaptation scheme.

### 3.1 Denoising Diffusion Models

Denoising diffusion models (Sohl-Dickstein et al., 2015; Ho et al., 2020) are latent variable generative models, with latent variables $\mathbf{x}_1, \mathbf{x}_2, ..., \mathbf{x}_T \in \mathbb{R}^n$ (the same dimensionality as the data $\mathbf{x} \sim q_{\mathbf{x}}$). Given a training sample $\mathbf{x}_0 \sim q_{\mathbf{x}}$, these models are based on constructing a diffusion process (forward process) of the variables $\mathbf{x}_{1:T} := \mathbf{x}_1, \mathbf{x}_2, ..., \mathbf{x}_T$ as a Markov chain from $\mathbf{x}_0$ to $\mathbf{x}_T$ of the form

$$q(\mathbf{x}_{1:T}|\mathbf{x}_0) := \prod_{t=1}^{T} q(\mathbf{x}_t|\mathbf{x}_{t-1}), \tag{2}$$

where $q(\mathbf{x}_t|\mathbf{x}_{t-1}) := \mathcal{N}(\sqrt{1 - \beta_t}\mathbf{x}_{t-1}, \beta_t\mathbf{I}_n)$, and $0 < \beta_1 < ... < \beta_T$ define the diffusion variance schedule (hyper-parameters of the model). Note that $\beta_T = 1$ ensures that $\mathbf{x}_T$ is pure Gaussian noise. Yet, as discussed in (Lin et al., 2024), this is not strictly enforced by many models. Note that sampling $\mathbf{x}_t|\mathbf{x}_0$ can be done via a simplified way using the parametrization (Ho et al., 2020):

$$\mathbf{x}_t = \sqrt{\bar{\alpha}_t}\mathbf{x}_0 + \sqrt{1 - \bar{\alpha}_t}\boldsymbol{\epsilon}, \ \boldsymbol{\epsilon} \sim \mathcal{N}(\mathbf{0}, \mathbf{I}_n), \tag{3}$$

where $\alpha_t := 1 - \beta_t$ and $\bar{\alpha}_t := \prod_{s=1}^{t} \alpha_s$. The goal of these models is to learn the distribution of the reverse chain from $\mathbf{x}_T$ to $\mathbf{x}_0$, which is parameterized as the Markov chain

$$p_\theta(\mathbf{x}_{0:T}) := p(\mathbf{x}_T) \prod_{t=1}^{T} p_\theta(\mathbf{x}_{t-1}|\mathbf{x}_t), \tag{4}$$

where $p_\theta(\mathbf{x}_{t-1}|\mathbf{x}_t) := \mathcal{N}(\boldsymbol{\mu}_\theta(\mathbf{x}_t, t), \boldsymbol{\Sigma}_\theta(\mathbf{x}_t, t))$,

$$\boldsymbol{\mu}_\theta(\mathbf{x}_t, t) := \frac{1}{\sqrt{\alpha_t}}\left(\mathbf{x}_t - \frac{1 - \alpha_t}{\sqrt{1 - \bar{\alpha}_t}} \boldsymbol{\epsilon}_\theta(\mathbf{x}_t, t)\right), \tag{5}$$

and $\theta$ denotes all the learnable parameters. Essentially, $\boldsymbol{\epsilon}_\theta(\mathbf{x}_t, t)$ is an estimator for the noise in $\mathbf{x}_t$ (up to scaling).

The parameters $\theta$ of the diffusion model $(\boldsymbol{\epsilon}_\theta(\mathbf{x}_t, t), \boldsymbol{\Sigma}_\theta(\mathbf{x}_t, t))$ are optimized by minimizing evidence lower bound (Sohl-Dickstein et al., 2015), a simplified score-matching loss (Ho et al., 2020; Song & Ermon, 2019), or a combination of both (Dhariwal & Nichol, 2021; Nichol & Dhariwal, 2021). For example, the simplified loss involves the minimization of

$$\ell_{\text{simple}}(\mathbf{x}_0, \boldsymbol{\epsilon}_\theta, t) = \|\boldsymbol{\epsilon} - \boldsymbol{\epsilon}_\theta(\sqrt{\bar{\alpha}_t}\mathbf{x}_0 + \sqrt{1 - \bar{\alpha}_t}\boldsymbol{\epsilon}, t)\|_2^2 \tag{6}$$

w.r.t. $\theta$ in each training iteration, where $\mathbf{x}_0$ is drawn from the training data, $t$ uniformly drawn from $\{1, ..., T\}$ and the noise $\boldsymbol{\epsilon} \sim \mathcal{N}(\mathbf{0}, \mathbf{I}_n)$.

Given a trained diffusion model $(\boldsymbol{\epsilon}_\theta(\mathbf{x}_t, t), \boldsymbol{\Sigma}_\theta(\mathbf{x}_t, t))$, one may generate a sample $\mathbf{x}_0$ from the learned data distribution $p_\theta$ by initializing $\mathbf{x}_T \sim \mathcal{N}(\mathbf{0}, \mathbf{I}_n)$ and running the reverse diffusion process by sampling

$$\mathbf{x}_{t-1} \sim \mathcal{N}(\boldsymbol{\mu}_\theta(\mathbf{x}_t, t), \boldsymbol{\Sigma}_\theta(\mathbf{x}_t, t)), \tag{7}$$

where $0 < t \leq T$ and $\boldsymbol{\mu}_\theta(\mathbf{x}_t, t)$ is defined in equation 5.

The class-guided sampling method that has been proposed in (Dhariwal & Nichol, 2021) modifies the sampling procedure in equation 7 by adding to the mean of the Gaussian a term that depends on the gradient of an offline-trained classifier, which has been trained using noisy images $\{\mathbf{x}_t\}$ for each $t$, and approximates the likelihood $p_{c|\mathbf{x}_t}$, where $c$ is the desired class. This procedure has been shown to improve the quality of the samples generated for the learned classes.

## 3.2  Diffusion based Image Reconstruction

We turn to extend the guidance method of (Dhariwal & Nichol, 2021) to image reconstruction. First, we generalize their framework to inverse problems in the form of equation 1. Namely, given the observed image $\mathbf{y}$, we modify the guided reverse diffusion process to generate possible reconstructions of $\mathbf{x}$ that are associated with $\mathbf{y}$ rather than arbitrary samples of a certain class. Similar to (Dhariwal & Nichol, 2021), ideally, the guiding direction at iteration $t$ should follow (the gradient of) the likelihood function $p_{\mathbf{y}|\mathbf{x}_t}$.

The key difference between our framework and (Dhariwal & Nichol, 2021) is that we need to base our method on the specific degraded image $\mathbf{y}$ rather than on a classifier that has been trained for each level of noise of $\{\mathbf{x}_t\}$. However, only the likelihood function $p_{\mathbf{y}|\mathbf{x}_0}$ is known, i.e., of the clean image $\mathbf{x}_0$ that is available only at the end of the procedure, and not for every $1 \leq t \leq T$. To overcome this issue, we propose a surrogate for the intermediate likelihood functions $p_{\mathbf{y}|\mathbf{x}_t}$. Our relaxation resembles the one in a recent concurrent work (Chung et al., 2022b). Yet, their sampling scheme is significantly different and has no adaptation ingredient.

Similar to (Dhariwal & Nichol, 2021), we guide the diffusion progression using the log-likelihood gradient. Formally, we are interested in sampling from the posterior

$$p_\theta(\mathbf{x}_t|\mathbf{x}_{t+1}, \mathbf{y}) \propto p_\theta(\mathbf{x}_t|\mathbf{x}_{t+1}) p_{\mathbf{y}|\mathbf{x}_t}(\mathbf{y}|\mathbf{x}_t), \tag{8}$$

where $p_{\mathbf{y}|\mathbf{x}_t}(\cdot|\mathbf{x}_t)$ is the distribution of $\mathbf{y}$ conditioned on $\mathbf{x}_t$, and $p_\theta(\mathbf{x}_t|\mathbf{x}_{t+1}) = \mathcal{N}(\boldsymbol{\mu}_\theta(\mathbf{x}_{t+1}, t+1)), \boldsymbol{\Sigma}_\theta(\mathbf{x}_{t+1}, t+1))$ is the learned diffusion prior. For brevity, we omit the arguments of $\boldsymbol{\mu}_\theta$ and $\boldsymbol{\Sigma}_\theta$ in the rest of this subsection.

Under the assumption that the likelihood $\log p_{\mathbf{y}|\mathbf{x}_t}(\mathbf{y}|\cdot)$ has low curvature compared to $\boldsymbol{\Sigma}_\theta^{-1}$ (Dhariwal & Nichol, 2021), the following Taylor expansion around $\mathbf{x}_t = \boldsymbol{\mu}_\theta$ is valid

$$\log p_{\mathbf{y}|\mathbf{x}_t}(\mathbf{y}|\mathbf{x}_t) \approx \log p_{\mathbf{y}|\mathbf{x}_t}(\mathbf{y}|\mathbf{x}_t)|_{\mathbf{x}_t=\boldsymbol{\mu}_\theta} + (\mathbf{x}_t - \boldsymbol{\mu}_\theta)^\top \nabla_{\mathbf{x}_t} \log p_{\mathbf{y}|\mathbf{x}_t}(\mathbf{y}|\mathbf{x}_t)|_{\mathbf{x}_t=\boldsymbol{\mu}_\theta} = (\mathbf{x}_t - \boldsymbol{\mu}_\theta)^\top \mathbf{g} + C_1, \quad (9)$$

where $\mathbf{g} = \nabla_{\mathbf{x}_t} \log p_{\mathbf{y}|\mathbf{x}_t}(\mathbf{y}|\mathbf{x}_t)|_{\mathbf{x}_t=\boldsymbol{\mu}_\theta}$, and $C_1$ is a constant that does not depend on $\mathbf{x}_t$. Then, similar to the computation in (Dhariwal & Nichol, 2021), we can use equation 9 to express the posterior in equation 8, i.e.,

$$\log(p_\theta(\mathbf{x}_t|\mathbf{x}_{t+1})p_{\mathbf{y}|\mathbf{x}_t}(\mathbf{y}|\mathbf{x}_t)) \approx C_2 + \log p(\mathbf{z}), \quad (10)$$

where $\mathbf{z} \sim \mathcal{N}(\boldsymbol{\mu}_\theta + \boldsymbol{\Sigma}_\theta \mathbf{g}, \boldsymbol{\Sigma}_\theta)$, and $C_2$ is some constant that does not depend on $\mathbf{x}_t$. Therefore, for conditioning the diffusion reverse process on $\mathbf{y}$, one needs to evaluate the derivative $\mathbf{g}$ from a (different) log-likelihood function $\log p_{\mathbf{y}|\mathbf{x}_t}(\mathbf{y}|\cdot)$ at each iteration $t$.

Observe that we know the exact log-likelihood function for $t = 0$. Since the noise $\mathbf{e}$ in equation 1 is white Gaussian with variance $\sigma^2$, we therefore have following distribution

$$p_{\mathbf{y}|\mathbf{x}}(\mathbf{y}|\mathbf{x}) = \mathcal{N}(\mathbf{A}\mathbf{x}, \sigma^2 \mathbf{I}_m) \propto e^{-\frac{1}{2\sigma^2}\|\mathbf{y}-\mathbf{A}\mathbf{x}\|_2^2}. \quad (11)$$

In the denoising diffusion setup, $\mathbf{y}$ is related to $\mathbf{x}_0$ using the observation model equation 1. Therefore,

$$\log p_{\mathbf{y}|\mathbf{x}_0}(\mathbf{y}|\mathbf{x}_0) \propto -\|\mathbf{A}\mathbf{x}_0 - \mathbf{y}\|_2^2. \quad (12)$$

However, we do not have tractable expressions for the likelihood functions $\{p_{\mathbf{y}|\mathbf{x}_t}(\mathbf{y}|\cdot)\}_{t=1}^T$. Therefore, motivated by the expression above, we propose the following approximation

$$\log p_{\mathbf{y}|\mathbf{x}_t}(\mathbf{y}|\mathbf{x}_t) \approx \log p_{\mathbf{y}|\mathbf{x}_0}(\mathbf{y}|\hat{\mathbf{x}}_0(\mathbf{x}_t)), \quad (13)$$

where

$$\hat{\mathbf{x}}_0(\mathbf{x}_t) := \left(\mathbf{x}_t - \sqrt{1-\bar{\alpha}_t}\boldsymbol{\epsilon}_\theta(\mathbf{x}_t, t)\right)/\sqrt{\bar{\alpha}_t} \quad (14)$$

is an estimation of $\mathbf{x}_0$ from $\mathbf{x}_t$, which is based on the (stochastic) relation of $\mathbf{x}_t$ and $\mathbf{x}_0$ in equation 3 and the random noise $\boldsymbol{\epsilon}$ is replaced by its estimation $\boldsymbol{\epsilon}_\theta(\mathbf{x}_t, t)$.

From equation 11 and equation 13 it follows that $\mathbf{g}$ in equation 9 can be approximated at each iteration $t$ by evaluating (e.g., via automatic-differentiation)

$$\mathbf{g} \approx -\nabla_{\mathbf{x}_t}\|\mathbf{A}\hat{\mathbf{x}}_0(\mathbf{x}_t) - \mathbf{y}\|_2^2|_{\mathbf{x}_t=\boldsymbol{\mu}_\theta}. \quad (15)$$

---

**Algorithm 1** Proposed GD sampling for image reconstruction given a diffusion model $(\boldsymbol{\epsilon}_\theta(\cdot), \boldsymbol{\Sigma}_\theta(\cdot))$, and a guidance scale $s$

**Require:** $(\boldsymbol{\epsilon}_\theta(\cdot), \boldsymbol{\Sigma}_\theta(\cdot))$, $\mathbf{y}$, $s$
1: $\mathbf{x}_T \leftarrow$ sample from $\mathcal{N}(\mathbf{0}, \mathbf{I}_n)$
2: **for** $t$ from $T$ to 1 **do**
3: $\quad \hat{\boldsymbol{\epsilon}}, \hat{\boldsymbol{\Sigma}} \leftarrow \boldsymbol{\epsilon}_\theta(\mathbf{x}_t, t), \boldsymbol{\Sigma}_\theta(\mathbf{x}_t, t)$
4: $\quad \hat{\boldsymbol{\mu}} \leftarrow \frac{1}{\sqrt{\alpha_t}}(\mathbf{x}_t - \frac{1-\alpha_t}{\sqrt{1-\bar{\alpha}_t}}\hat{\boldsymbol{\epsilon}})$
5: $\quad \mathbf{y}_t \leftarrow \sqrt{\bar{\alpha}_t}\mathbf{y} + \sqrt{1-\bar{\alpha}_t}\mathbf{A}\hat{\boldsymbol{\epsilon}}$
6: $\quad \mathbf{g} \leftarrow -2\mathbf{A}^T(\mathbf{A}\hat{\boldsymbol{\mu}} - \mathbf{y}_t)$
7: $\quad \mathbf{x}_{t-1} \leftarrow$ sample from $\mathcal{N}(\hat{\boldsymbol{\mu}} + s\hat{\boldsymbol{\Sigma}}\mathbf{g}, \hat{\boldsymbol{\Sigma}})$
8: **end forreturn** $\mathbf{x}_0$

---

Note that existing methods (Chung et al., 2022b; Kawar et al., 2022a; Song et al., 2021) either use a term that resembles equation 15 with the naive approximation $\hat{\mathbf{x}}_0(\mathbf{x}_t) = \mathbf{x}_t$ (Kawar et al., 2022a; Song et al., 2021), or significantly modify equation 15 before computing it via the automatic derivation framework (Chung et al., 2022b) (we observed that trying to compute the exact equation 15 is unstable due to numerical issues). For example, in the official implementation

of (Chung et al., 2022b), which uses automatic derivation, the squaring of the norm in equation 15 is dropped even though this is not stated in their paper (otherwise, the reconstruction suffers from significant artifacts). In our case, we use the following relaxation to overcome the stability issue of using equation 15 directly. For a pretrained denoiser predicting $\boldsymbol{\epsilon}_\theta$ from $\mathbf{x}_t$ and $0 < t \le T$ we have

$$
\begin{aligned}
\|\mathbf{A}\hat{\mathbf{x}}_0(\mathbf{x}_t) - \mathbf{y}\|_2^2 &= \|\mathbf{A}(\mathbf{x}_t - \sqrt{1-\bar{\alpha}_t}\boldsymbol{\epsilon}_\theta)/\sqrt{\bar{\alpha}_t} - \mathbf{y}\|_2^2 \\
&\propto \|\mathbf{A}\mathbf{x}_t - \sqrt{1-\bar{\alpha}_t}\mathbf{A}\boldsymbol{\epsilon}_\theta - \sqrt{\bar{\alpha}_t}\mathbf{y}\|_2^2 \\
&= \|\mathbf{A}\mathbf{x}_t - \sqrt{\bar{\alpha}_t}\mathbf{y} - \sqrt{1-\bar{\alpha}_t}\mathbf{A}\boldsymbol{\epsilon}_\theta\|_2^2 \\
&= \|\mathbf{A}\mathbf{x}_t - \mathbf{y}_t\|_2^2,
\end{aligned}
\tag{16}
$$

where $\mathbf{y}_t := \sqrt{\bar{\alpha}_t}\mathbf{y} + \sqrt{1-\bar{\alpha}_t}\mathbf{A}\boldsymbol{\epsilon}_\theta$. We further assume that $\boldsymbol{\epsilon}_\theta$ is independent of $\mathbf{x}_t$, which we found to be sufficient in our use-cases. Consequently, we propose to replace the expression for $\mathbf{g}$ (the guiding likelihood direction at each iteration $t$) that is given in equation 15 with a surrogate obtained by evaluating the derivative of equation 16 w.r.t. $\mathbf{x}_t$, which is given by

$$
\mathbf{g} = -(2\mathbf{A}^T(\mathbf{A}\mathbf{x}_t - \mathbf{y}_t) - 2\nabla_{\mathbf{x}_t}\mathbf{y}_t(\mathbf{A}\mathbf{x}_t - \mathbf{y}_t))m|_{\mathbf{x}_t = \boldsymbol{\mu}_\theta} \approx -2\mathbf{A}^T(\mathbf{A}\mathbf{x}_t - \mathbf{y}_t)|_{\mathbf{x}_t = \boldsymbol{\mu}_\theta}
\tag{17}
$$

that can be used for sampling the posterior distribution as detailed in Algorithm 1.

## 3.3 Adaptive Diffusion

Having defined the guided inverse diffusion flow for image reconstruction, we turn to discuss how one may adapt a given diffusion model to a given degraded image $\mathbf{y}$ as defined in equation 1. Assume we have a pretrained diffusion model $(\boldsymbol{\epsilon}_\theta(\cdot), \boldsymbol{\Sigma}_\theta(\cdot))$, then the adaptation scheme is defined by the following minimization problem

$$
\hat{\theta} = \arg\min_\theta \sum_{t=1}^T \ell_{\text{simple}}(\mathbf{y}, \boldsymbol{\epsilon}_\theta, t)
\tag{18}
$$

with $\ell_{\text{simple}}$ defined in equation 6, which can be solved using stochastic gradient descent, where at each iteration the gradient step is performed on a single term of the sum above, for $0 < t \le T$ chosen randomly. Although the original work (Dhariwal & Nichol, 2021) trains the network to predict the posterior variance $\Sigma_\theta$, in our case, we did not see any benefit of including it in the adaptation loss.

Adapting the denoising network to the measurement image $\mathbf{y}$, allows it to learn cross-scale features recurring in the image, which is a well studied property of natural images (Ulyanov et al., 2018; Mataev et al., 2019; Shaham et al., 2019; Michaeli & Irani, 2014). Such an approach has been proven to be very helpful in reconstruction-based algorithms (Hussein et al., 2020a; Tirer & Giryes, 2019). However, in some cases where the image does not satisfy the assumption of recurring patterns across scales, this approach can lose some of the sharpness captured in training. Therefore, in this work we extend the approach to few-shot fine-tuning adaptation, where instead of solving equation 18 w.r.t. $\mathbf{y}$, we propose an algorithm for retrieving $K$ images similar to $\mathbf{x}$ from a large dataset of diverse images, using off-the-shelf embedding distance.

Let $(\xi_v(\cdot), \xi_\ell(\cdot))$ be some off-the-shelf multi-modal encoder trained on visual-language modalities, e.g., CLIP (Radford et al., 2021), BLIP (Li et al., 2022b), or CyCLIP (Goel et al., 2022)). Let $\xi_v(\cdot)$ and $\xi_\ell(\cdot)$ be the visual and language encoders respectively. Then, given a large diverse dataset of natural images, we propose to retrieve $K$ images, denoted by $\{\mathbf{z}_k\}_{k=1}^K$, with minimal embedding distance from $\mathbf{y}$. Formally, let $\mathcal{D}_{\text{IA}}$ be an arbitrary external dataset, then

$$
\begin{aligned}
\{\mathbf{z}_k\}_{k=1}^K = \{\mathbf{z}_1, ..., \mathbf{z}_K | \phi_\xi(\mathbf{z}_1, \mathbf{y}) \le ... &\le \phi_\xi(\mathbf{z}_K, \mathbf{y}) \\
&\le \phi_\xi(\mathbf{z}, \mathbf{y}), \forall \mathbf{z} \in \mathcal{D}_{\text{IA}} \setminus \{\mathbf{z}_1, ..., \mathbf{z}_K\}\},
\end{aligned}
\tag{19}
$$

where $\phi_\xi(\mathbf{a}, \mathbf{b}) = 2\arcsin(0.5\|\xi(\mathbf{a}) - \xi(\mathbf{b})\|_2^2)$ is the spherical distance and $\xi$ can be either the visual or language encoder depending on the provided conditioning of the application.

After retrieving $K$-NN images $\{\mathbf{z}_k\}_{k=1}^K$ from $\mathcal{D}_{\text{IA}}$, we fine-tune the diffusion model on them, which adapts the denoising network to the context of $\mathbf{y}$. Specifically, we modify the denoiser parameters $\theta$ based on minimizing a loss similar

to equation 18, but with $\{\mathbf{z}_k\}_{k=1}^{K}$ rather than $\mathbf{y}$. We stochastically solve the following minimization problem

$$\hat{\theta} = \arg \min_{\theta} \sum_{k=1}^{K} \sum_{t=1}^{T} \ell_{\text{simple}}(\mathbf{z}_k, \boldsymbol{\epsilon}_{\theta}, t) \tag{20}$$

We refer to this K-NN based adaptation technique as ADIR (Adaptive Diffusion for Image Reconstruction), which is described schematically in Figure 2.

| | IA Iter. | LR | NN imag. | $s$ | diff. steps |
|---|---|---|---|---|---|
| ADIR-GD | 400 | $10^{-4}$ | 20 | 10 | 1000 |
| ADIR-SD | 400 | $10^{-4}$ | 50 | - | 50 |

Table 1: Configurations used for ADIR.

## 4 Experiments

We evaluate our method on two state-of-the-art diffusion models, Guided Diffusion (GD) (Dhariwal & Nichol, 2021) and Stable Diffusion (SD) (Rombach et al., 2022), showing results for super-resolution, colorization and deblurring. In addition, we show how adaptive diffusion can be used for the task of text-based editing using stable diffusion.

Guided diffusion (Dhariwal & Nichol, 2021) provides several models with a conditioning mechanism built-in to the denoiser. However, in our case, we perform the conditioning using the $\log$-likelihood term. Therefore, we used the unconditional model that was trained on ImageNet (Russakovsky et al., 2015) and produces images of size $256 \times 256$. In the original work, the conditioning for generating an image from an arbitrary class was performed using a classifier trained to classify the noisy sample $\mathbf{x}_t$ directly, where the log-likelihood derivative can be obtained by deriving the corresponding logits w.r.t. $\mathbf{x}_t$ directly. In our setup, the conditioning is performed using $\mathbf{g}$ in equation 17, where $\mathbf{A}$ is defined by the reconstruction task, which we specify in the sequel.

In addition to GD, we demonstrate the improvement that can be achieved using stable diffusion (Rombach et al., 2022), where we use publicly available super-resolution and text-based editing models for it. Instead of training the denoiser on the natural images domain directly, they suggest using a Variational Auto Encoder (VAE) and train the denoiser using a latent representation of the data. Note that the lower dimensionality of the latent enables the network to be trained at higher resolutions.

In all cases, we adapt the diffusion models in the image adaptive scheme presented in section 3.3, using the Google Open Dataset (Kuznetsova et al., 2020) as the external dataset $\mathcal{D}_{\text{IA}}$, from which we retrieve $K$ images, where $K = 20$ for GD and $K = 50$ for SD (several examples of retrieved images are shown Figure 21). Since the Nearest Neighbor (NN) search is performed in the embedding space, we can efficiently retrieve the $K$ images from the $1.7M$ images using a K-D Tree structure. This significantly accelerates the retrieval procedure, as can be seen in Table 6. We compare the reconstruction performance and the runtimes when using random NN images, MSE-based NN, and using our approach. Because the MSE-based retrieval uses the whole image for the search, applying the K-D Tree structure for such a scheme is more challenging. Instead, one may use random projection in order to decrease the representation dimension, however, because no benefits were obtained from such retrieval without the projection, we did not explore such a direction.

For optimizing the network parameters we use LoRA (Hu et al., 2021) with rank $r = 16$ and scaling $\alpha = 8$ for all the convolution layers, which is then optimized using Adam (Kingma & Ba, 2014). The specific implementation configurations are detailed in Table 1. We run all of our experiments on a NVIDIA RTX A6000 48GB card, which allows us to fine-tune the models by randomly sampling a batch of 6 images from $\{\mathbf{z}_k\}_{k=1}^{K}$, where in each iteration we use the same $0 < t \le T$ for images in the batch.

### 4.1 Super Resolution

In the Super-Resolution (SR) task one would like to reconstruct a high resolution image $\mathbf{x}$ from its low resolution image $\mathbf{y}$, where in this case $\mathbf{A}$ represents an anti-aliasing filter followed by sub-sampling with stride $\gamma$, which we refer

|  | SRx4 | SRx8 |
|---|---|---|
| IPT | 0.237/5.02/64.19 | - |
| USRNet | 0.249/4.38/45.76 | - |
| SwinIR | 0.232/4.98/64.19 | 0.424/**4.54**/48.04 |
| SRDiff | **0.135**/4.76/60.87 | - |
| Real-ESRGAN | 0.317/5.02/**69.42** | - |
| DeepRED | 0.475/3.20/22.77 | 0.591/2.99/17.43 |
| DDRM | 0.297/3.42/28.96 | 0.572/3.13/20.68 |
| GD | 0.325/4.88/64.63 | 0.365/4.36/53.99 |
| ADIR | 0.335/**5.06**/66.33 | **0.347**/4.41/**55.89** |

Table 2: x4 Super resolution results ($128^2 \rightarrow 512^2$) and 8 ($64^2 \rightarrow 512^2$) for the unconditional guided diffusion model (Dhariwal & Nichol, 2021). The results are averaged on the first 50 images of the DIV2K validation set (Agustsson & Timofte, 2017). We compare ADIR to IPT (Chen et al., 2021), USR-Net (Zhang et al., 2020), SwinIR (Liang et al., 2021), SRDiff (Li et al., 2022a), Real-ESRGAN (Wang et al., 2021), DeepRED (Mataev et al., 2019), the baseline approach presented in Section 3.2(without adaptation), and DDRM (Kawar et al., 2022a). We use the traditional LPIPS (Zhang et al., 2018) as well as the state-of-the-art no reference perceptual losses AVA-MUSIQ and KonIQ-MUSIQ (Ke et al., 2021) for evaluation (LPIPS/MUSIQ-AVA/MUSIQ-KONIQ). The best results are in bold black, and the second best is highlighted in blue.

|  | SRx4 |
|---|---|
| IPT | 0.221/4.90/65.38 |
| USRNet | 0.234/4.51/59.10 |
| SwinIR | 0.218/4.88/65.08 |
| SRDiff | 0.237/4.76/62.64 |
| DeepRED | 0.405/3.25/25.26 |
| Real-ESRGAN | 0.305/4.93/69.11 |
| Stable Diffusion | 0.331/5.07/69.18 |
| ADIR (SD) | **0.213/5.51/72.56** |

Table 3: x4 Super resolution ($256^2 \rightarrow 1024^2$) using Stable Diffusion SR (Rombach et al., 2022). Similar to Table 2, the results are averaged on the first 50 images of the DIV2K validation set (Agustsson & Timofte, 2017). We compare ADIR to IPT (Chen et al., 2021), US-RNet (Zhang et al., 2020), SwinIR (Liang et al., 2021), SRDiff (Li et al., 2022a), DeepRED (Mataev et al., 2019), Stable Diffusion (without adaptation), and Real-ESRGAN (Wang et al., 2021). We use LPIPS (Zhang et al., 2018) as well as AVA-MUSIQ and KonIQ-MUSIQ (Ke et al., 2021) for evaluation (LPIPS/MUSIQ-AVA/MUSIQ-KONIQ). The best results are in bold black, and the second best is highlighted in blue.

|  | Box (256) | Box (512) | Gauss (256) |
|---|---|---|---|
| M3SNet | 0.477/3.13/26.16 | 0.468/2.93/47.42 | 0.481/2.75/31.13 |
| DeepRED | 0.561/3.61/22.12 | 0.557/3.57/27.59 | 0.572/3.59/19.13 |
| Restormer | **0.341**/3.74/40.11 | 0.377/4.67/55.03 | 0.518/3.61/36.70 |
| MPRNet | 0.395/3.08/26.90 | 0.429/3.63/37.87 | 0.491/3.01/20.96 |
| Guided Diffusion | 0.423/4.20/49.19 | 0.411/**4.81**/58.66 | 0.424/4.01/48.11 |
| ADIR (GD) | 0.394/**4.31**/**55.78** | **0.312**/4.77/**60.13** | 0.415/**4.19**/**51.80** |

Table 4: Deblurring with 10 noise levels results for the unconditional guided diffusion model (Dhariwal & Nichol, 2021). Similar to SR in Table 2, the results are averaged on the first 50 images of the DIV2K validation set (Agustsson & Timofte, 2017). We compare our method to M3SNet (Gao et al., 2023), DeepRED (Mataev et al., 2019), Restormer (Zamir et al., 2022), MPRNet Mehri et al. (2021) and the baseline presented in Section 3.2 (without adaptation). We use LPIPS (Zhang et al., 2018) as well as AVA-MUSIQ and KonIQ-MUSIQ (Ke et al., 2021) for evaluation (LPIPS/MUSIQ-AVA/MUSIQ-KONIQ).

to as the scaling factor. In our use-case we employ a bicubic anti-aliasing filter and assume $\mathbf{e} = 0$, similarly to most SR works.

Here we apply our approach on two different diffusion based SR methods, Stable Diffusion (Rombach et al., 2022), and section 3.2 approach combined with the unconditional diffusion model from (Dhariwal & Nichol, 2021). In Stable Diffusion, the low-resolution image $\mathbf{y}$ is upscaled from $256 \times 256$ to $1024 \times 1024$, while in Guided Diffusion we use the unconditional model trained on $256 \times 256$ images. When adapting Stable diffusion, we downsample random crops of the $K$-NN images using $\mathbf{A}$, which we encode using the VAE and plug into the network conditioning mechanism.

| DDRM | Guided Diffusion (GD) | ADIR (GD) |
|---|---|---|
| 4.012/53.458 | 4.195/56.044 | **4.214/58.679** |

Table 5: Image colorization for the unconditional guided diffusion model (Dhariwal & Nichol, 2021). The results are averaged on the first 50 images of the DIV2K validation set (Agustsson & Timofte, 2017). We compare ADIR to the baseline presented in Section 3.2 (without adaptation) and DDRM (Kawar et al., 2022a). We use AVA-MUSIQ and KonIQ-MUSIQ (Ke et al., 2021) for evaluation (MUSIQ-AVA/MUSIQ-KONIQ). The best results are in bold black, and the second best is highlighted in blue.

We fine-tune both models using random crops of the $K$-NN images, to which we then add noise using the scheduler provided by each model.

The perception preference of generative models-based image reconstruction has been seen in many works (Hussein et al., 2020a; Bora et al., 2017; Blau & Michaeli, 2018). Therefore, we chose a perception-based measure to evaluate the performance of our method. Specifically, we use the state-of-the-art AVA-MUSIQ and KonIQ-MUSIQ perceptual quality assessment measures (Ke et al., 2021), which are state-of-the-art image quality assessment measures. We report our results using the two measures averaged on the first 50 validation images of the DIV2K (Agustsson & Timofte, 2017) dataset. As can be seen in Tables 2, 3, our method significantly outperforms both Stable Diffusion and GD-based reconstruction approaches. We compare our super-resolution (SR) results to Stable Diffusion SR and Guided Diffusion without the adaptation component. Additionally, we benchmark our method against several other state-of-the-art techniques; IPT (Chen et al., 2021), USRNet (Zhang et al., 2020), SwinIR (Liang et al., 2021), SRDiff (Li et al., 2022a), DeepRED (Mataev et al., 2019), Stable Diffusion (without adaptation), and Real-ESRGAN (Wang et al., 2021). As can be seen from the results, our method outperforms or shows competitive results compared to the other approaches. It is worth noting that because ADIR is a generative prior-based method, it targets the perceptual quality aspect more than the distortional aspect; a fact can be seen when comparing ADIR to task-specific approaches using reference-based measures (e.g. LPIPS). We also compare ADIR to adaptation using random images from the dataset, as well as MSE-based retrieval, and report the results in Table 6, as well as Figure 6, where the obvious advantage of ADIR can be seen clearly.

Figures 1 and 3 present qualitative results. Note that our method achieves superior restoration quality. In some cases it restores even fine details that were blurred in the acquisition of the GT image.

## 4.2 Deblurring

In deblurring, $\mathbf{y}$ is obtained by applying a blur filter (uniform blur of size $5 \times 5$ in our case) on $\mathbf{x}$, followed by adding measurement noise $\mathbf{e} \sim \mathcal{N}(0, \sigma^2 I_n)$, where in our setting $\sigma = 10$. We apply our proposed approach in Section 3.2 for the Guided Diffusion unconditional model (Dhariwal & Nichol, 2021) to solve the task.

As a baseline, we use the unconditional diffusion model provided by GD (Dhariwal & Nichol, 2021), which was trained on $256 \times 256$ size images. Yet, in our tests, we solve the deblurring task on images of sizes $256 \times 256$ and $512 \times 512$, which emphasizes the remarkable benefit of the adaptation, as it allows the model to generalize to resolutions not seen during training.

Similar to SR, in Table 4 we report the KonIQ-MUSIQ and AVA-MUSIQ (Ke et al., 2021) measures, averaged on the first 50 DIV2K validation images (Agustsson & Timofte, 2017), where we compare our approach to the guided diffusion reconstruction without image adaptation. Visual comparisons are also available in Figure 5, where a significant improvement can be seen in both robustness to noise and reconstructing details. We also compare ADIR to the scenario where we adapt the denoiser on random images from the dataset, as well as MSE-based retrieval; as can be seen in Table 6 and Figure 6.

## 4.3 Colorization

In colorization, $\mathbf{y}$ is obtained by averaging the colors of $\mathbf{x}$ using RGB2Gray transform. Similar to deblurring, we apply our proposed approach in Section 3.2 to solve the task. In this case, $\mathbf{A}$ can be implemented by averaging the color dimension of $\mathbf{x}$, while $\mathbf{A}^T$ can simply be viewed as a replication of the color dimension. We use the unconditional

| | SRx8 | | Deblur | |
|---|---|---|---|---|
| | LPIPS/AVA/KONIQ | runtime [sec/image] | LPIPS/AVA/KONIQ | runtime [sec/image] |
| GD (baseline) | 0.365/4.36/53.99 | **830** | 0.423/4.20/49.19 | **425** |
| IA w/ random images | 0.430/4.14/52.89 | 1300 | 0.433/3.68/40.05 | 895 |
| IA w/ MSE based NN | 0.434/4.28/53.12 | 2700 | 0.428/3.62/39.13 | 2500 |
| ADIR | **0.347/4.41/55.89** | 1308 | **0.394/4.31/55.78** | 903 |

Table 6: Ablation study of the adaptation advantage of ADIR. We compare our method to adaptation using random images sampled from the dataset, adaptation using MSE as a retrieval distance, and the approach from Section 3.2. We use the traditional LPIPS (Zhang et al., 2018) as well as the state-of-the-art no reference perceptual losses AVA-MUSIQ and KonIQ-MUSIQ (Ke et al., 2021) for evaluation. We also benchmark the runtime of each method and report the inference time per image on a single NVIDIA RTX A6000 GPU. The best results are in bold black, and the second best is highlighted in blue. In ADIR we achieve much smaller runtimes compared to the MSE-based retrieval because we use an efficient K-D tree data structure on the embedding space that is much smaller than the naive pixel domain.

| Original Image | Masked | Stable Diffusion | Stable Diffusion | ADIR | ADIR |
|---|---|---|---|---|---|

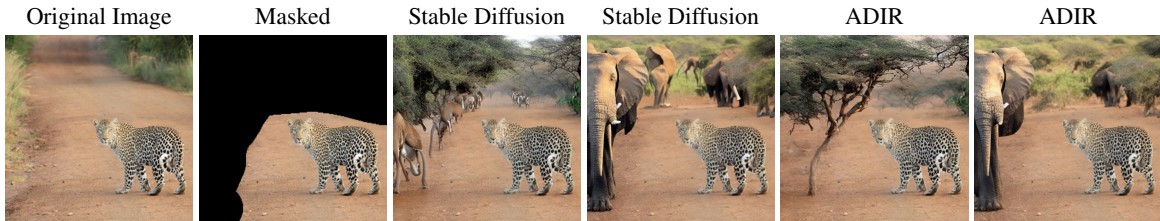

Figure 4: Text-based editing comparison between Stable Diffusion and ADIR, using the prompt "Africa" for two different seeds. Note that Stable diffusion adds partial animals while ADIR completes the scene more naturally.

diffusion model provided by GD (Dhariwal & Nichol, 2021) as a baseline for coloring $256 \times 256$ images. Visual comparison of the results can be seen in Figure 7. We report the average MUSIQ (Ke et al., 2021) perceptual measure for this case, as shown in Table 5. Note that we do not report LPIPS as there are many colorization solutions and therefore the reconstructed image may differ a lot from the ground truth. Thus, we focus on non-reference based perceptual measures for the colorization task.

## 4.4 Text-Guided Editing

Text-guided image editing is the task of completing a masked region of $\mathbf{x}$ according to a prompt provided by the user. In this case, the diffusion model needs to predict objects and textures correspondent to the provided prompt, therefore we chose to adapt the network on $\{\mathbf{z}_k\}_{k=1}^K$ retrieved using the text encoder, i.e. by solving equation 19 using $\xi_\ell$. For evaluating our method for this application, we use the inpainting model of Stable Diffusion (Rombach et al., 2022). Where $\mathbf{y}$ encoded and concatenated with the mask resized to latent dimension, which are then plugged to the denoising network. When adapting the network, we follow the training scheme of Stable Diffusion, where we use random masks and the classifier-free conditioning approach (Ho & Salimans, 2022) used for training Stable Diffusion, where the text embedding is randomly chosen to either be the encoded prompt or the embedding of an empty prompt. Notice that we cannot compare to (Giannone et al., 2022; Sheynin et al., 2022; Kawar et al., 2022b) as there is no code available for them. For some of them, we do not even have access to the diffusion model that they adapt (Saharia et al., 2022a). Note though that our goal is not to show state-of-the-art editing results but rather to show here the potential contribution of ADIR to text-guided editing. As it is a general framework, it may be used also with other existing editing techniques in order to improve them.

Figure 19 presents the editing results and compares them to both stable diffusion and GLIDE. GLIDE is the basis of the popular DALL-E-2 model. The images of GLIDE are taken from the paper. We use ADIR with stable diffusion and optimize them using the same seed.

Since Stable Diffusion was trained using a lossy latent representation with smaller dimensionality than the data, it is clear that GLIDE can achieve better results. However, because our method adapts the network to a specific scenario,

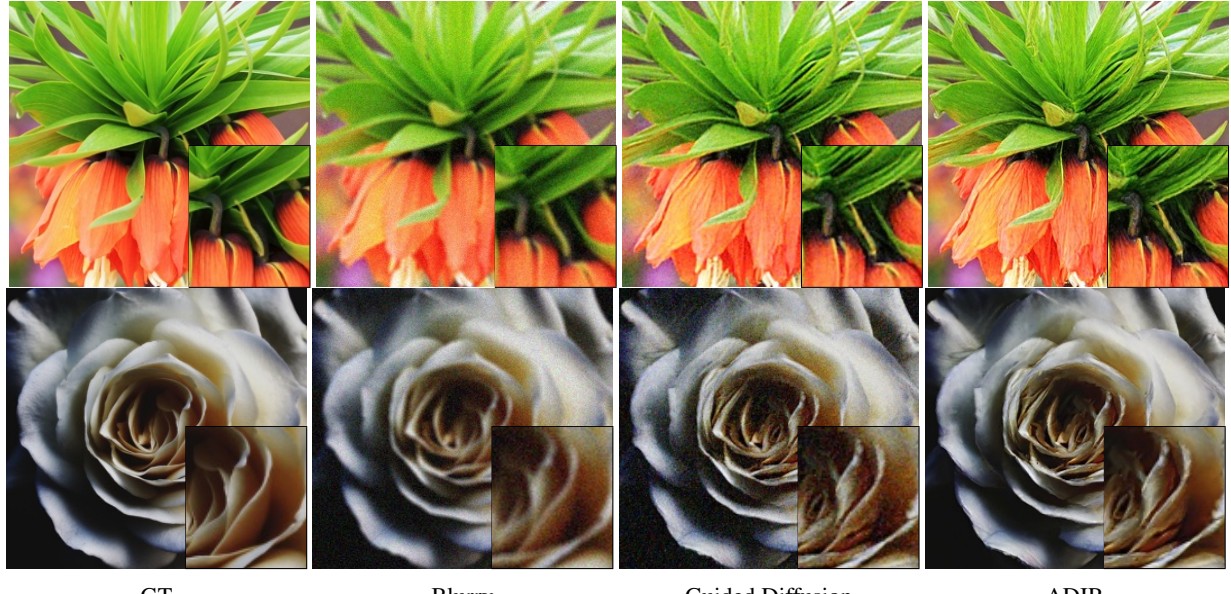

GT                Blurry            Guided Diffusion           ADIR

Figure 5: Image deblurring using Guided Diffusion approach from section 3.2 and ADIR, using the unconditional model from (Dhariwal & Nichol, 2021). The degradation is performed using $5 \times 5$ uniform blur filter with 10 levels of additive Gaussian noise. Note the better quality of our method.

it enables the model to produce cleaner and more accurate generations, as can be seen in Figure 19. In the first image we see that Stable Diffusion adds an object that does not blend well and has artifacts, while when combined with our approach the quality improves significantly. Similarly, in the second image we see that Stable Diffusion produces an inaccurate edit, where it adds a brown hair instead of red hair. This is again improved by our adaptation method.

**Limitation.** One limitation of our approach is that as is the case with all diffusion models, there is randomness in the generation process of the results. Therefore, the quality of the output may depend on the random seed being used. For a fair comparison, we used the same seed both for ADIR and the baseline. In the appendix, we provide more examples with different random seeds. We still find that when we compare our approach and the baseline with the same seed, we consistently get an improvement. Another limitation of ADIR is that it works sequentially, i.e. we first look for $K$-NN images and then fine-tune the denoiser network on these images, therefore, an additional run-time is added to the standard diffusion flow. Also, in this work, we assume that the observation operator $\mathbf{A}$ is known (non-blind setting), while in many real-world applications, it is usually inaccessible. As a result, one needs to run the guidance scheme (section 3.2) with an estimated version of $\mathbf{A}$, which is suboptimal. Additionally, for optimal performance, one should use a relatively diverse dataset to retrieve images that match the degraded image context. Otherwise, the adaptation can lead to negligible advantage. We leave exploring these questions to a future research.

## 5 Conclusion

We have presented the Adaptive Diffusion Image Reconstruction (ADIR) method, in which we improve the reconstruction results in several imaging tasks using off-the-shelf diffusion models. We have demonstrated how our adaptation can significantly improve existing state-of-the-art methods, e.g. Stable Diffusion for super resolution, where the exploitation of external data with the same context as $\mathbf{y}$, combined with our adaptation scheme leads to a significant improvement. Specifically, the produced images are sharper and have more details than the original ground truth image. Importantly, note that our novel adaptive diffusion ingredient can be incorporated into any conditional sampling scheme that is based on diffusion models, beyond those that are examined in this paper. One such possible direction is integrating our method with advanced diffusion models-based editing techniques (Meng et al., 2022; Kim et al., 2022; Mokady et al., 2023; Bar-Tal et al., 2023; Molad et al., 2023; Wei et al., 2023; Huang et al., 2023; Qi et al., 2023; Liu et al., 2023). Yet, we believe that our proposed novel concept can be a useful tool for improving diffusion-based reconstruction and editing.

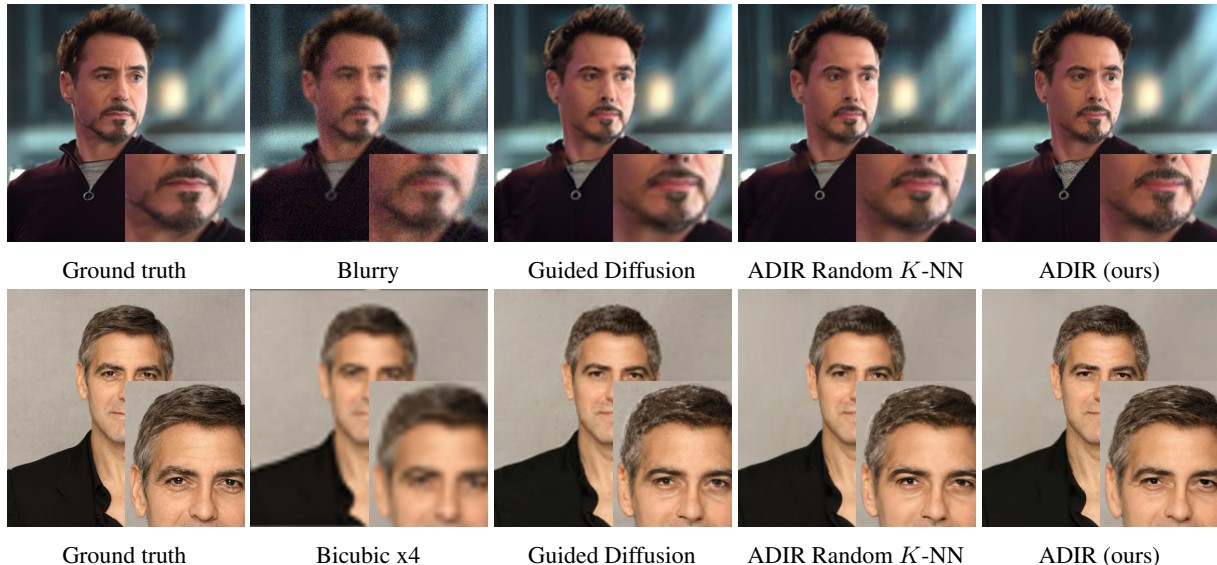

| Ground truth | Blurry | Guided Diffusion | ADIR Random $K$-NN | ADIR (ours) |

| Ground truth | Bicubic x4 | Guided Diffusion | ADIR Random $K$-NN | ADIR (ours) |

Figure 6: Ablation study on the benefit of ADIR compared to adapting the denoiser on random images for deblurring (upper row) and super-resolution (bottom row) of celebrity images. We compare ADIR applied using random images of the celebrity from the web, to random NN images, and guided diffusion with no adaptation from section 3.2.

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

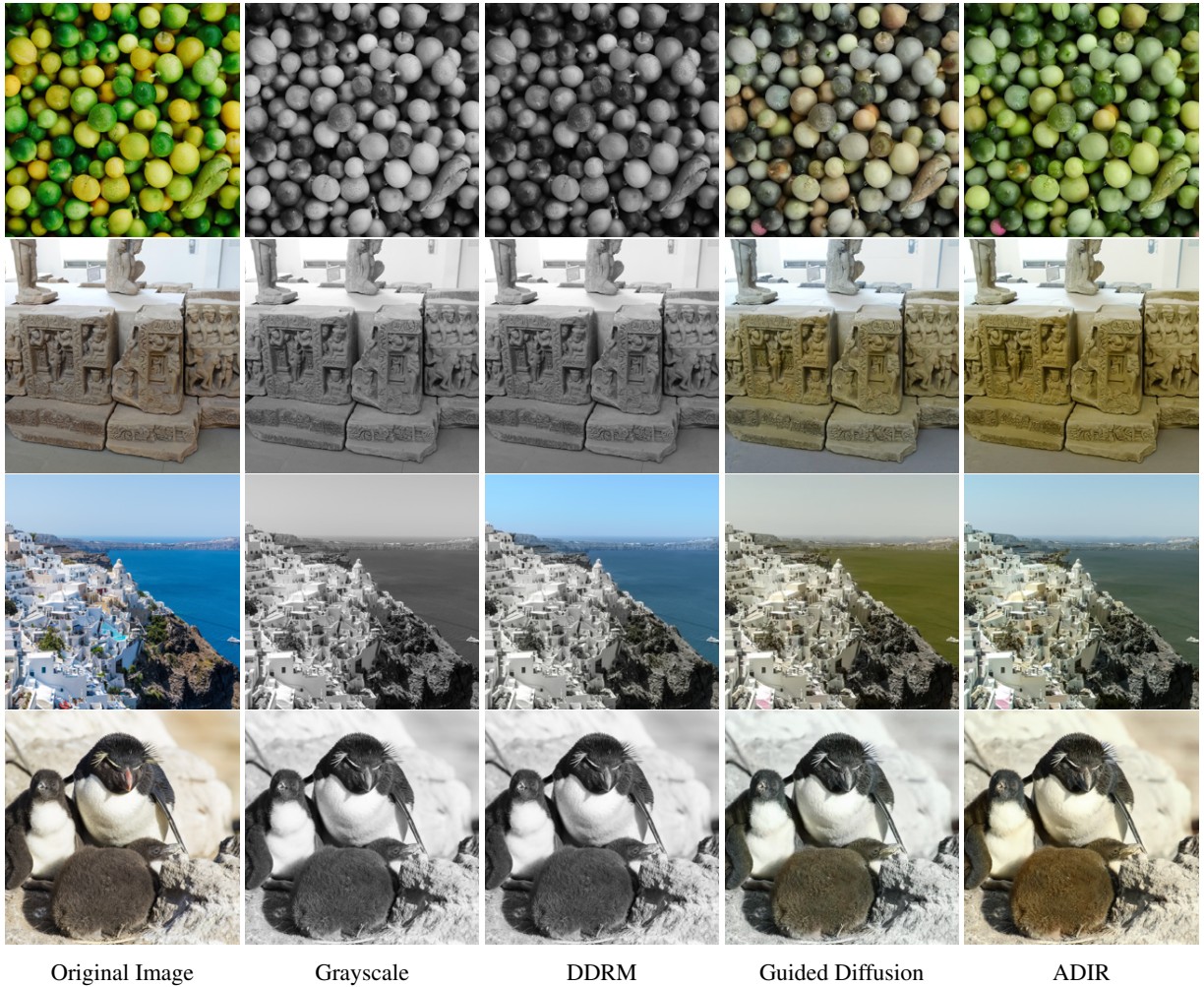

|  |  |  |  |  |
|---|---|---|---|---|
| Original Image | Grayscale | DDRM | Guided Diffusion | ADIR |

Figure 7: Image colorization results comparison between DDRM (Kawar et al., 2022a), Guided diffusion proposed in section 3.2, and our adaptive approach ADIR. As can be seen, adapting the denoiser network to the given image can improve the results significantly.

Jooyoung Choi, Sungwon Kim, Yonghyun Jeong, Youngjune Gwon, and Sungroh Yoon. Ilvr: Conditioning method for denoising diffusion probabilistic models. In *2021 IEEE/CVF International Conference on Computer Vision (ICCV)*, pp. 14347–14356. IEEE, 2021.

Hyungjin Chung, Eun Sun Lee, and Jong Chul Ye. Mr image denoising and super-resolution using regularized reverse diffusion. *arXiv preprint arXiv:2203.12621*, 2022a.

Hyungjin Chung, Byeongsu Sim, Dohoon Ryu, and Jong Chul Ye. Improving diffusion models for inverse problems using manifold constraints. *arXiv preprint arXiv:2206.00941*, 2022b.

Mauricio Delbracio and Peyman Milanfar. Inversion by direct iteration: An alternative to denoising diffusion for image restoration. *arXiv preprint arXiv:2303.11435*, 2023.

Manik Dhar, Aditya Grover, and Stefano Ermon. Modeling sparse deviations for compressed sensing using generative models. In *International Conference on Machine Learning*, pp. 1214–1223. PMLR, 2018.

Prafulla Dhariwal and Alexander Nichol. Diffusion models beat GANs on image synthesis. *Advances in Neural Information Processing Systems*, 34:8780–8794, 2021.

Chao Dong, Chen Change Loy, Kaiming He, and Xiaoou Tang. Image super-resolution using deep convolutional networks. *IEEE transactions on pattern analysis and machine intelligence*, 38(2):295–307, 2015.

Yossi Gandelsman, Yu Sun, Xinlei Chen, and Alexei A Efros. Test-time training with masked autoencoders. *arXiv preprint arXiv:2209.07522*, 2022.

Hu Gao, Jing Yang, Ying Zhang, Ning Wang, Jingfan Yang, and Depeng Dang. A mountain-shaped single-stage network for accurate image restoration. *arXiv preprint arXiv:2305.05146*, 2023.

Tomer Garber and Tom Tirer. Image restoration by denoising diffusion models with iteratively preconditioned guidance. *arXiv preprint arXiv:2312.16519*, 2023.

Giorgio Giannone, Didrik Nielsen, and Ole Winther. Few-shot diffusion models. *10.48550/ARXIV.2205.15463*, 2022.

Shashank Goel, Hritik Bansal, Sumit Bhatia, Ryan A Rossi, Vishwa Vinay, and Aditya Grover. Cyclip: Cyclic contrastive language-image pretraining. *arXiv preprint arXiv:2205.14459*, 2022.

Jonathan Ho and Tim Salimans. Classifier-free diffusion guidance. *arXiv preprint arXiv:2207.12598*, 2022.

Jonathan Ho, Ajay Jain, and Pieter Abbeel. Denoising diffusion probabilistic models. *Advances in Neural Information Processing Systems*, 33:6840–6851, 2020.

Jonathan Ho, Tim Salimans, Alexey Gritsenko, William Chan, Mohammad Norouzi, and David J Fleet. Video diffusion models. *arXiv:2204.03458*, 2022.

Edward J Hu, Yelong Shen, Phillip Wallis, Zeyuan Allen-Zhu, Yuanzhi Li, Shean Wang, Lu Wang, and Weizhu Chen. Lora: Low-rank adaptation of large language models. *arXiv preprint arXiv:2106.09685*, 2021.

Lianghua Huang, Di Chen, Yu Liu, Shen Yujun, Deli Zhao, and Zhou Jingren. Composer: Creative and controllable image synthesis with composable conditions. 2023.

Shady Abu Hussein, Tom Tirer, and Raja Giryes. Image-adaptive gan based reconstruction. In *Proceedings of the AAAI Conference on Artificial Intelligence*, volume 34, pp. 3121–3129, 2020a.

Shady Abu Hussein, Tom Tirer, and Raja Giryes. Correction filter for single image super-resolution: Robustifying off-the-shelf deep super-resolvers. In *Proceedings of the IEEE/CVF Conference on Computer Vision and Pattern Recognition*, pp. 1428–1437, 2020b.

Xiaozhong Ji, Yun Cao, Ying Tai, Chengjie Wang, Jilin Li, and Feiyue Huang. Real-world super-resolution via kernel estimation and noise injection. In *IEEE/CVF Conference on Computer Vision and Pattern Recognition Workshops (CVPRW)*, pp. 1914–1923, 2020.

Bahjat Kawar, Michael Elad, Stefano Ermon, and Jiaming Song. Denoising diffusion restoration models. *arXiv preprint arXiv:2201.11793*, 2022a.

Bahjat Kawar, Shiran Zada, Oran Lang, Omer Tov, Huiwen Chang, Tali Dekel, Inbar Mosseri, and Michal Irani. Imagic: Text-based real image editing with diffusion models. *arXiv preprint arXiv:2210.09276*, 2022b.

Junjie Ke, Qifei Wang, Yilin Wang, Peyman Milanfar, and Feng Yang. Musiq: Multi-scale image quality transformer. In *Proceedings of the IEEE/CVF International Conference on Computer Vision*, pp. 5148–5157, 2021.

Gwanghyun Kim, Taesung Kwon, and Jong Chul Ye. Diffusionclip: Text-guided diffusion models for robust image manipulation. In *Proceedings of the IEEE/CVF Conference on Computer Vision and Pattern Recognition (CVPR)*, pp. 2426–2435, June 2022.

Diederik P Kingma and Jimmy Ba. Adam: A method for stochastic optimization. *arXiv preprint arXiv:1412.6980*, 2014.

Alina Kuznetsova, Hassan Rom, Neil Alldrin, Jasper Uijlings, Ivan Krasin, Jordi Pont-Tuset, Shahab Kamali, Stefan Popov, Matteo Malloci, Alexander Kolesnikov, Tom Duerig, and Vittorio Ferrari. The open images dataset v4: Unified image classification, object detection, and visual relationship detection at scale. *IJCV*, 2020.

Haoying Li, Yifan Yang, Meng Chang, Shiqi Chen, Huajun Feng, Zhihai Xu, Qi Li, and Yueting Chen. Srdiff: Single image super-resolution with diffusion probabilistic models. *Neurocomputing*, 479:47–59, 2022a.

Junnan Li, Dongxu Li, Caiming Xiong, and Steven Hoi. Blip: Bootstrapping language-image pre-training for unified vision-language understanding and generation. *arXiv preprint arXiv:2201.12086*, 2022b.

Jingyun Liang, Jiezhang Cao, Guolei Sun, Kai Zhang, Luc Van Gool, and Radu Timofte. Swinir: Image restoration using swin transformer. In *Proceedings of the IEEE/CVF International Conference on Computer Vision (ICCV) Workshops*, pp. 1833–1844, October 2021.

Bee Lim, Sanghyun Son, Heewon Kim, Seungjun Nah, and Kyoung Mu Lee. Enhanced deep residual networks for single image super-resolution. In *Proceedings of the IEEE conference on computer vision and pattern recognition workshops*, pp. 136–144, 2017.

Shanchuan Lin, Bingchen Liu, Jiashi Li, and Xiao Yang. Common diffusion noise schedules and sample steps are flawed. In *Proceedings of the IEEE/CVF Winter Conference on Applications of Computer Vision*, pp. 5404–5411, 2024.

Shaoteng Liu, Yuechen Zhang, Wenbo Li, Zhe Lin, and Jiaya Jia. Video-p2p: Video editing with cross-attention control. *arXiv:2303.04761*, 2023.

Andreas Lugmayr, Martin Danelljan, Luc Van Gool, and Radu Timofte. Srflow: Learning the super-resolution space with normalizing flow. In *ECCV*, 2020.

Andreas Lugmayr, Martin Danelljan, Andres Romero, Fisher Yu, Radu Timofte, and Luc Van Gool. Repaint: In-painting using denoising diffusion probabilistic models. In *Proceedings of the IEEE/CVF Conference on Computer Vision and Pattern Recognition*, pp. 11461–11471, 2022.

Gary Mataev, Peyman Milanfar, and Michael Elad. Deepred: Deep image prior powered by red. In *Proceedings of the IEEE/CVF International Conference on Computer Vision Workshops*, pp. 0–0, 2019.

Armin Mehri, Parichehr B Ardakani, and Angel D Sappa. Mprnet: Multi-path residual network for lightweight image super resolution. In *Proceedings of the IEEE/CVF Winter Conference on Applications of Computer Vision*, pp. 2704–2713, 2021.

Chenlin Meng, Yutong He, Yang Song, Jiaming Song, Jiajun Wu, Jun-Yan Zhu, and Stefano Ermon. SDEdit: Guided image synthesis and editing with stochastic differential equations. In *International Conference on Learning Representations*, 2022.

Tomer Michaeli and Michal Irani. Blind deblurring using internal patch recurrence. In *Computer Vision–ECCV 2014: 13th European Conference, Zurich, Switzerland, September 6-12, 2014, Proceedings, Part III 13*, pp. 783–798. Springer, 2014.

Ron Mokady, Amir Hertz, Kfir Aberman, Yael Pritch, and Daniel Cohen-Or. Null-text inversion for editing real images using guided diffusion models. In *CVPR*, 2023.

Eyal Molad, Eliahu Horwitz, Dani Valevski, Alex Rav Acha, Yossi Matias, Yael Pritch, Yaniv Leviathan, and Yedid Hoshen. Dreamix: Video diffusion models are general video editors. *arXiv preprint arXiv:2302.01329*, 2023.

Alex Nichol, Prafulla Dhariwal, Aditya Ramesh, Pranav Shyam, Pamela Mishkin, Bob McGrew, Ilya Sutskever, and Mark Chen. Glide: Towards photorealistic image generation and editing with text-guided diffusion models. *arXiv preprint arXiv:2112.10741*, 2021.

Alexander Quinn Nichol and Prafulla Dhariwal. Improved denoising diffusion probabilistic models. In *International Conference on Machine Learning*, pp. 8162–8171. PMLR, 2021.

Yotam Nitzan, Kfir Aberman, Qiurui He, Orly Liba, Michal Yarom, Yossi Gandelsman, Inbar Mosseri, Yael Pritch, and Daniel Cohen-Or. Mystyle: A personalized generative prior. *arXiv preprint arXiv:2203.17272*, 2022.

Ozan Özdenizci and Robert Legenstein. Restoring vision in adverse weather conditions with patch-based denoising diffusion models. *IEEE Transactions on Pattern Analysis and Machine Intelligence*, 2023.

Xingang Pan, Xiaohang Zhan, Bo Dai, Dahua Lin, Chen Change Loy, and Ping Luo. Exploiting deep generative prior for versatile image restoration and manipulation. *IEEE Transactions on Pattern Analysis and Machine Intelligence*, 2021.

Chenyang Qi, Xiaodong Cun, Yong Zhang, Chenyang Lei, Xintao Wang, Ying Shan, and Qifeng Chen. Fatezero: Fusing attentions for zero-shot text-based video editing. *arXiv:2303.09535*, 2023.

Alec Radford, Jong Wook Kim, Chris Hallacy, Aditya Ramesh, Gabriel Goh, Sandhini Agarwal, Girish Sastry, Amanda Askell, Pamela Mishkin, Jack Clark, et al. Learning transferable visual models from natural language supervision. In *International Conference on Machine Learning*, pp. 8748–8763. PMLR, 2021.

Daniel Roich, Ron Mokady, Amit H Bermano, and Daniel Cohen-Or. Pivotal tuning for latent-based editing of real images. *ACM Transactions on Graphics (TOG)*, 42(1):1–13, 2022.

Robin Rombach, Andreas Blattmann, Dominik Lorenz, Patrick Esser, and Björn Ommer. High-resolution image synthesis with latent diffusion models. In *Proceedings of the IEEE/CVF Conference on Computer Vision and Pattern Recognition*, pp. 10684–10695, 2022.

Olga Russakovsky, Jia Deng, Hao Su, Jonathan Krause, Sanjeev Satheesh, Sean Ma, Zhiheng Huang, Andrej Karpathy, Aditya Khosla, Michael Bernstein, et al. Imagenet large scale visual recognition challenge. *International journal of computer vision*, 115(3):211–252, 2015.

Chitwan Saharia, William Chan, Saurabh Saxena, Lala Li, Jay Whang, Emily Denton, Seyed Kamyar Seyed Ghasemipour, Burcu Karagol Ayan, S. Sara Mahdavi, Rapha Gontijo Lopes, Tim Salimans, Jonathan Ho, David J Fleet, and Mohammad Norouzi. Photorealistic text-to-image diffusion models with deep language understanding. *arXiv:2205.11487*, 2022a.

Chitwan Saharia, Jonathan Ho, William Chan, Tim Salimans, David J Fleet, and Mohammad Norouzi. Image super-resolution via iterative refinement. *IEEE Transactions on Pattern Analysis and Machine Intelligence*, 2022b.

Tamar Rott Shaham, Tali Dekel, and Tomer Michaeli. Singan: Learning a generative model from a single natural image. In *Proceedings of the IEEE/CVF international conference on computer vision*, pp. 4570–4580, 2019.

Shelly Sheynin, Oron Ashual, Adam Polyak, Uriel Singer, Oran Gafni, Eliya Nachmani, and Yaniv Taigman. Knn-diffusion: Image generation via large-scale retrieval. *arXiv:2204.02849*, 2022.

Assaf Shocher, Nadav Cohen, and Michal Irani. "zero-shot" super-resolution using deep internal learning. In *Proceedings of the IEEE Conference on Computer Vision and Pattern Recognition (CVPR)*, pp. 3118–3126, 2018.

Jascha Sohl-Dickstein, Eric Weiss, Niru Maheswaranathan, and Surya Ganguli. Deep unsupervised learning using nonequilibrium thermodynamics. In *International Conference on Machine Learning*, pp. 2256–2265. PMLR, 2015.

Yang Song and Stefano Ermon. Generative modeling by estimating gradients of the data distribution. *Advances in Neural Information Processing Systems*, 32, 2019.

Yang Song, Liyue Shen, Lei Xing, and Stefano Ermon. Solving inverse problems in medical imaging with score-based generative models. *arXiv preprint arXiv:2111.08005*, 2021.

Jian Sun, Wenfei Cao, Zongben Xu, and Jean Ponce. Learning a convolutional neural network for non-uniform motion blur removal. *2015 IEEE Conference on Computer Vision and Pattern Recognition (CVPR)*, pp. 769–777, 2015.

Tom Tirer and Raja Giryes. Image restoration by iterative denoising and backward projections. *IEEE Transactions on Image Processing*, 28(3):1220–1234, 2018.

Tom Tirer and Raja Giryes. Super-resolution via image-adapted denoising cnns: Incorporating external and internal learning. *IEEE Signal Processing Letters*, 26(7):1080–1084, 2019.

Dmitry Ulyanov, Andrea Vedaldi, and Victor Lempitsky. Deep image prior. In *Proceedings of the IEEE Conference on Computer Vision and Pattern Recognition (CVPR)*, pp. 9446–9454, 2018.

Xintao Wang, Liangbin Xie, Chao Dong, and Ying Shan. Real-esrgan: Training real-world blind super-resolution with pure synthetic data. In *Proceedings of the IEEE/CVF International Conference on Computer Vision*, pp. 1905–1914, 2021.

Pengxu Wei, Ziwei Xie, Hannan Lu, ZongYuan Zhan, Qixiang Ye, Wangmeng Zuo, and Liang Lin. Component divide-and-conquer for real-world image super-resolution. In *Proceedings of the European Conference on Computer Vision*, 2020.

Yuxiang Wei, Yabo Zhang, Zhilong Ji, Jinfeng Bai, Lei Zhang, and Wangmeng Zuo. Elite: Encoding visual concepts into textual embeddings for customized text-to-image generation. *arXiv preprint arXiv:2302.13848*, 2023.

Jay Whang, Mauricio Delbracio, Hossein Talebi, Chitwan Saharia, Alexandros G Dimakis, and Peyman Milanfar. Deblurring via stochastic refinement. In *Proceedings of the IEEE/CVF Conference on Computer Vision and Pattern Recognition*, pp. 16293–16303, 2022.

Syed Waqas Zamir, Aditya Arora, Salman Khan, Munawar Hayat, Fahad Shahbaz Khan, and Ming-Hsuan Yang. Restormer: Efficient transformer for high-resolution image restoration. In *Proceedings of the IEEE/CVF conference on computer vision and pattern recognition*, pp. 5728–5739, 2022.

Kai Zhang, Wangmeng Zuo, Yunjin Chen, Deyu Meng, and Lei Zhang. Beyond a Gaussian denoiser: Residual learning of deep cnn for image denoising. *IEEE Transactions on Image Processing*, 26(7):3142–3155, 2017a.

Kai Zhang, Wangmeng Zuo, Shuhang Gu, and Lei Zhang. Learning deep cnn denoiser prior for image restoration. In *Proceedings of the IEEE conference on computer vision and pattern recognition*, pp. 3929–3938, 2017b.

Kai Zhang, Luc Van Gool, and Radu Timofte. Deep unfolding network for image super-resolution. In *Proceedings of the IEEE/CVF conference on computer vision and pattern recognition*, pp. 3217–3226, 2020.

Kai Zhang, Yawei Li, Wangmeng Zuo, Lei Zhang, Luc Van Gool, and Radu Timofte. Plug-and-play image restoration with deep denoiser prior. *IEEE Transactions on Pattern Analysis and Machine Intelligence*, 2021a.

Kai Zhang, Jingyun Liang, Luc Van Gool, and Radu Timofte. Designing a practical degradation model for deep blind image super-resolution. In *Proceedings of the IEEE/CVF International Conference on Computer Vision (ICCV)*, pp. 4791–4800, October 2021b.

Kai Zhang, Yawei Li, Jingyun Liang, Jiezhang Cao, Yulun Zhang, Hao Tang, Radu Timofte, and Luc Van Gool. Practical blind denoising via swin-conv-unet and data synthesis. *arXiv preprint*, 2022.

Richard Zhang, Phillip Isola, Alexei A Efros, Eli Shechtman, and Oliver Wang. The unreasonable effectiveness of deep features as a perceptual metric. In *Proceedings of the IEEE conference on computer vision and pattern recognition*, pp. 586–595, 2018.

Yuanzhi Zhu, Kai Zhang, Jingyun Liang, Jiezhang Cao, Bihan Wen, Radu Timofte, and Luc Van Gool. Denoising diffusion models for plug-and-play image restoration. In *Proceedings of the IEEE/CVF Conference on Computer Vision and Pattern Recognition*, pp. 1219–1229, 2023.

## Additional Results

In the following we

- Show results for super resolution with scaling factor of 8.

- Show additional results of deblurring task.

- Show more results of colorization use-case.

- Compare our method to Stable Diffusion for editing task in multiple scenarios.

- Examples of retrieved nearest neighbours images (Figure 21).

- Examine the effect of $\mathbf{A}$ on the $K$-NN retrieval (Figure 22).

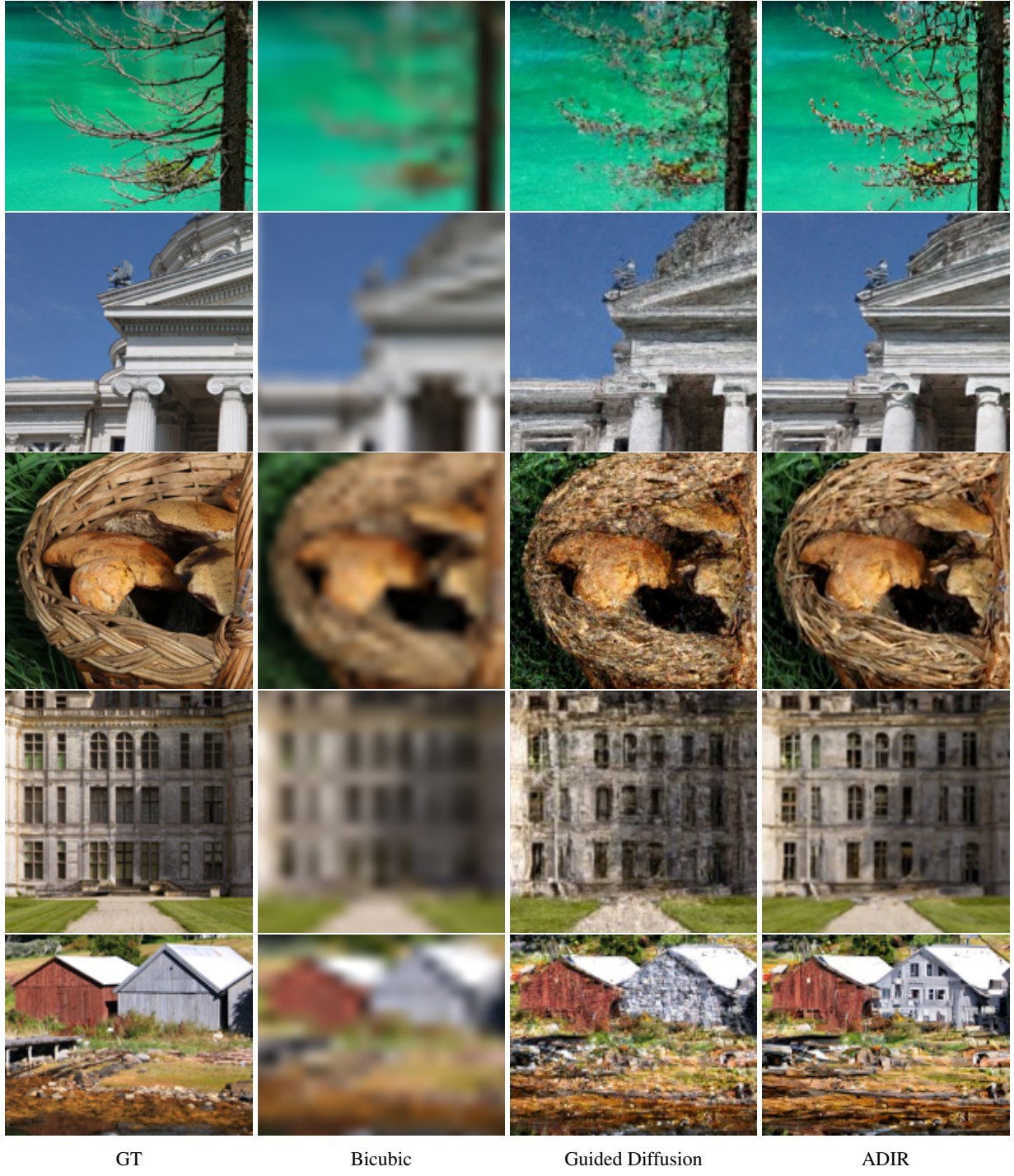

| GT | Bicubic | Guided Diffusion | ADIR |

Figure 8: Comparison of super resolution ($64^2 \to 512^2$) results of Guided Diffusion from section 3.2 and our method (ADIR), using the unconditional model from (Rombach et al., 2022). As can be seen from the images, our method outperforms guided diffusion in both sharpness and reconstruction details.

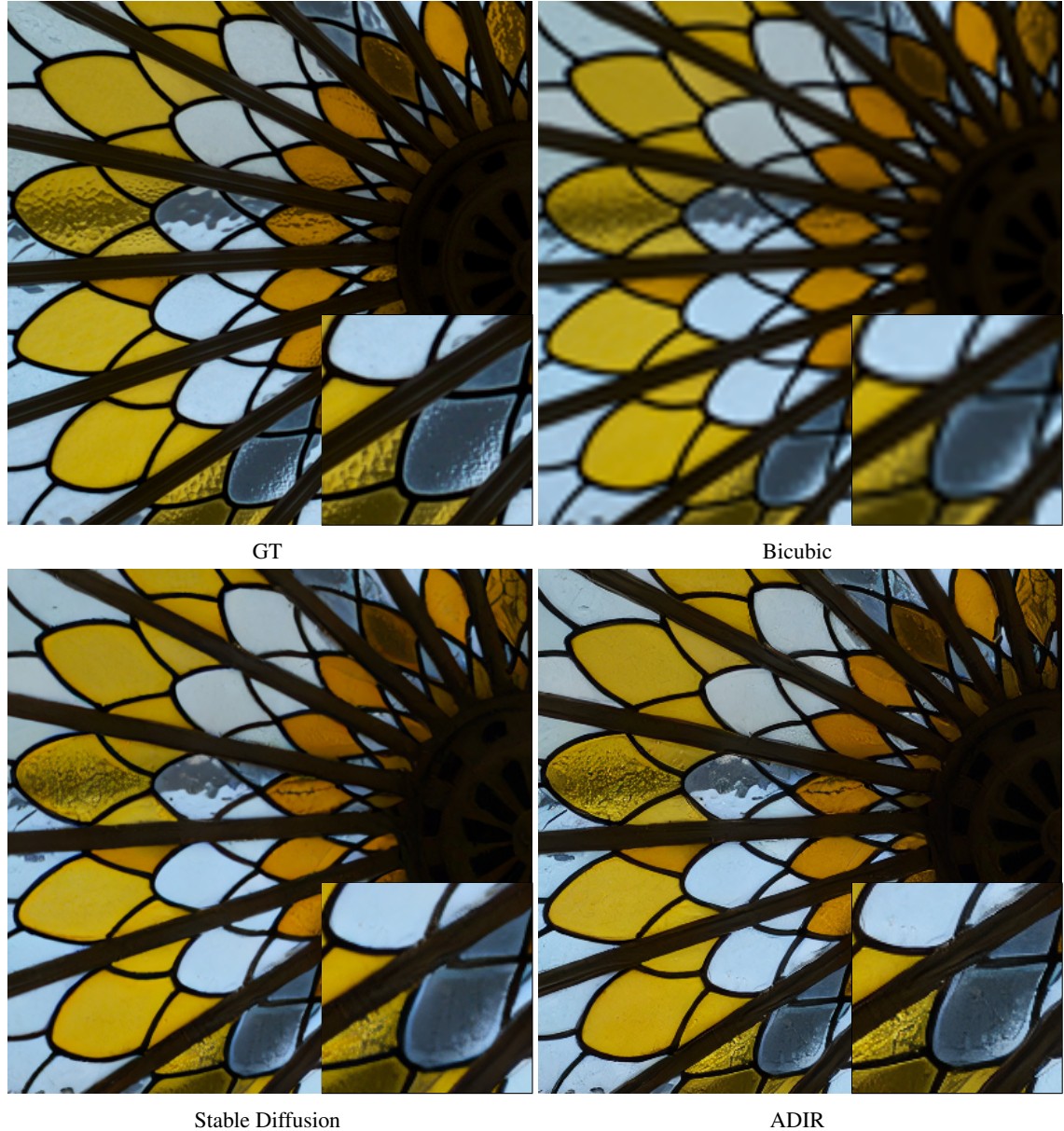

Figure 9: Comparison of super resolution ($256^2 \rightarrow 1024^2$) results of Stable Diffusion (Rombach et al., 2022) and our method (ADIR), using the unconditional model from (Rombach et al., 2022). As can be seen from the images, our method outperforms guided diffusion in both sharpness and reconstruction details.

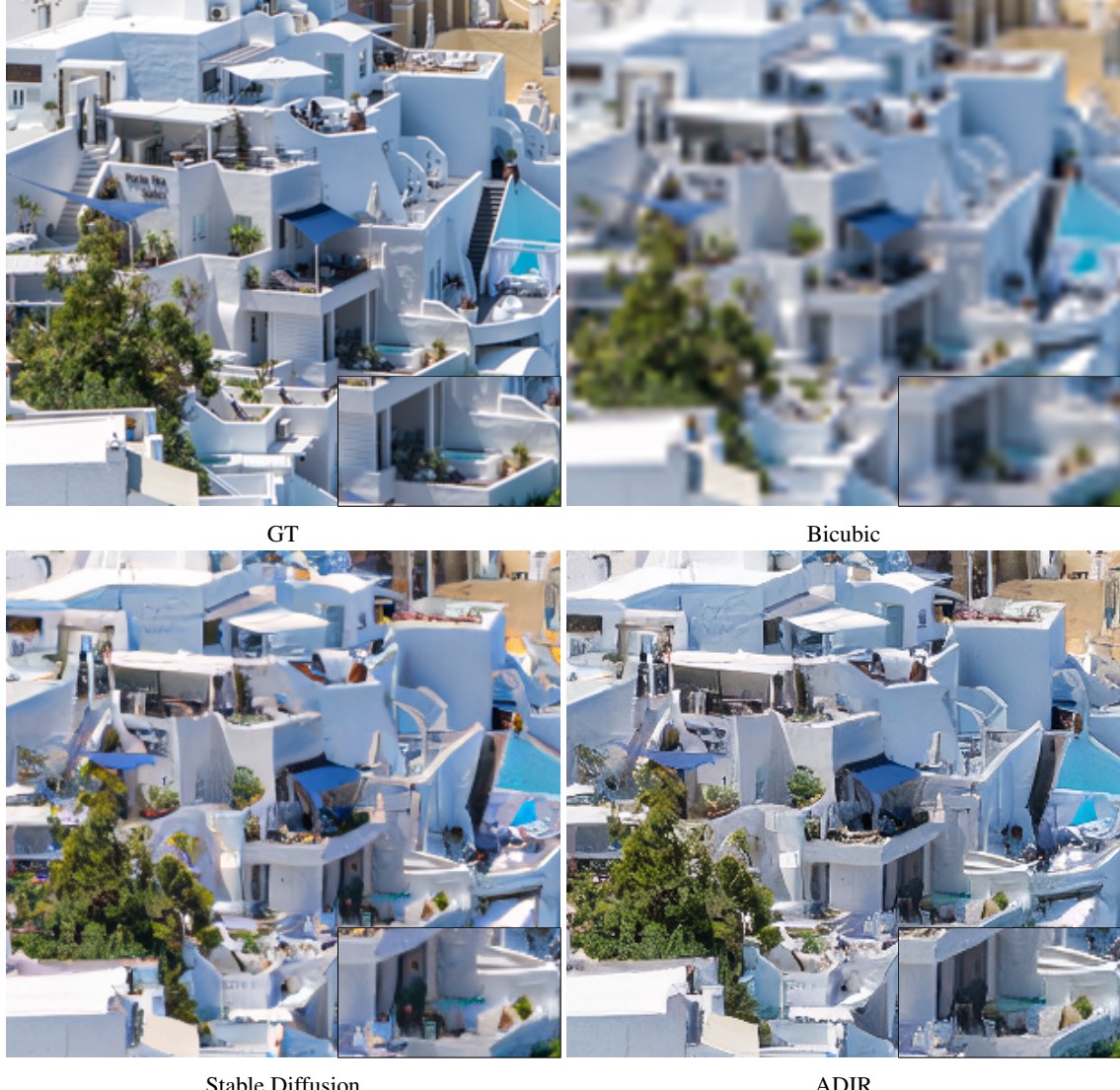

GT

Bicubic

Stable Diffusion

ADIR

Figure 10: Comparison of super resolution ($256^2 \rightarrow 1024^2$) results of Stable Diffusion (Rombach et al., 2022) and our method (ADIR), using the unconditional model from (Rombach et al., 2022). As can be seen from the images, our method outperforms guided diffusion in both sharpness and reconstruction details.

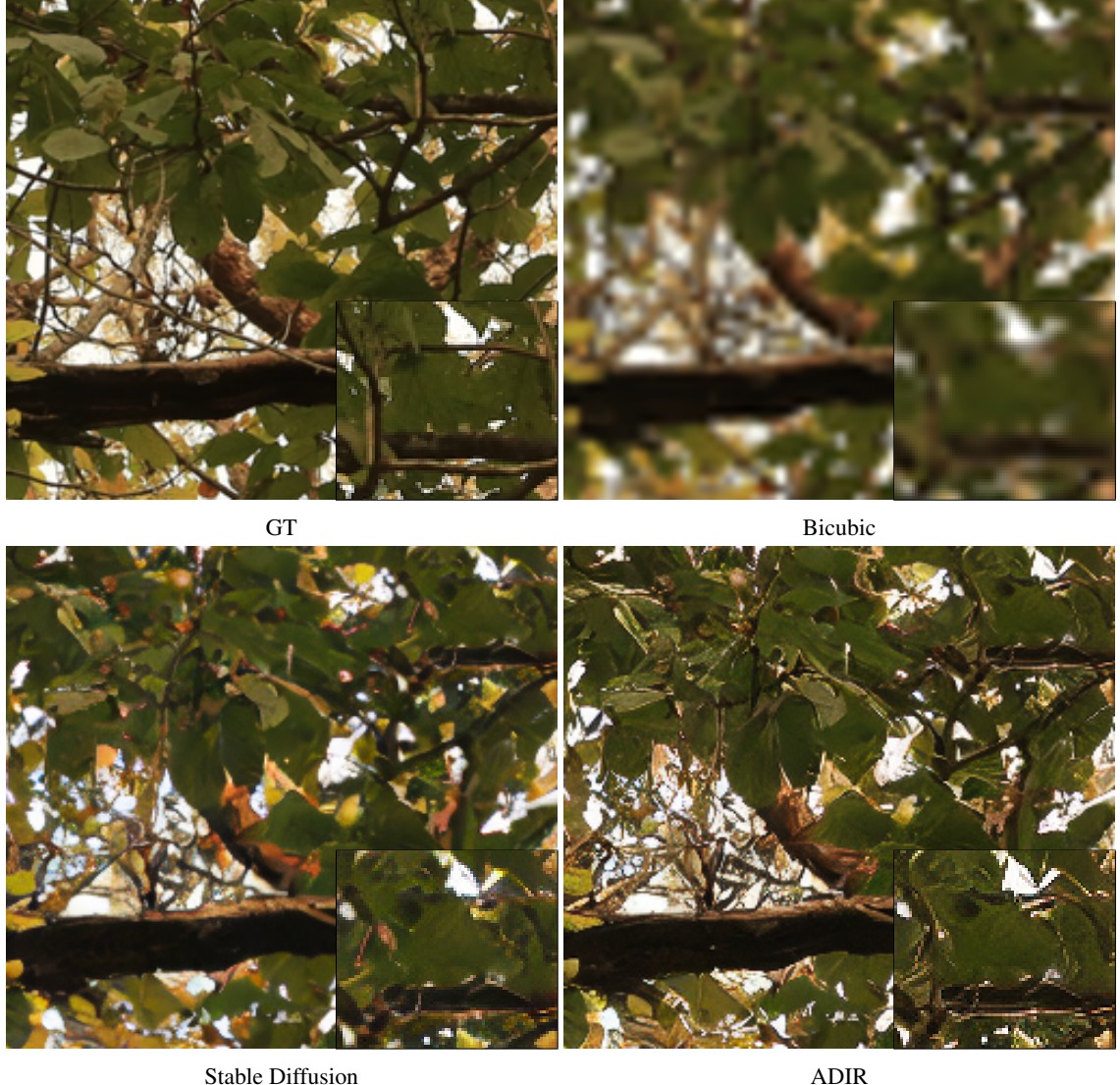

GT                                          Bicubic

Stable Diffusion                            ADIR

Figure 11: Comparison of super resolution ($256^2 \rightarrow 1024^2$) results of Stable Diffusion (Rombach et al., 2022) and our method (ADIR), using the unconditional model from (Rombach et al., 2022). As can be seen from the images, our method outperforms guided diffusion in both sharpness and reconstruction details.

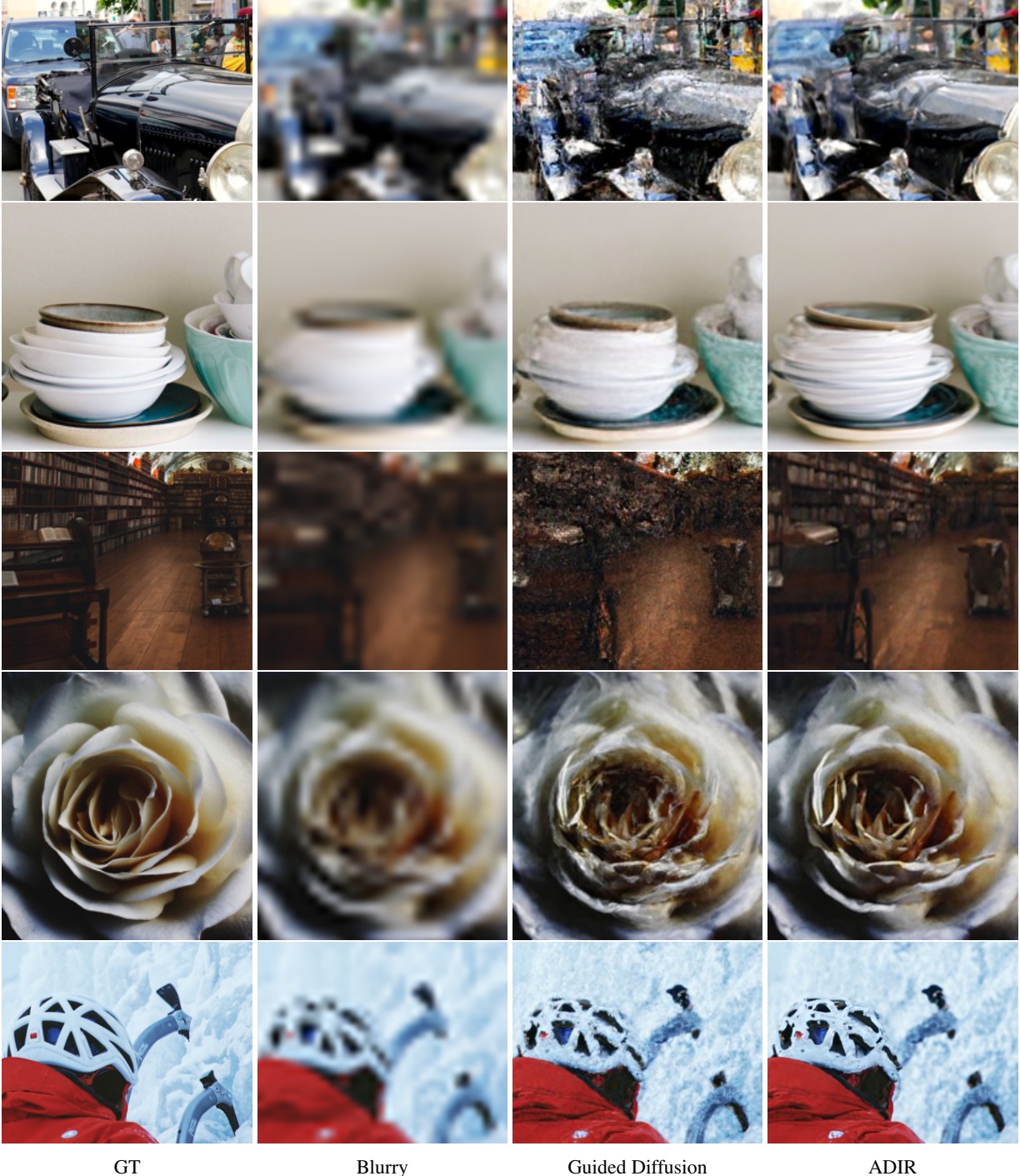

GT       Blurry     Guided Diffusion    ADIR

Figure 12: Comparison of super resolution ($64^2 \rightarrow 512^2$) results of Guided Diffusion from section 3.2 and our method (ADIR), using the unconditional model from (Rombach et al., 2022). As can be seen from the images, our method outperforms guided diffusion in both sharpness and reconstruction details.

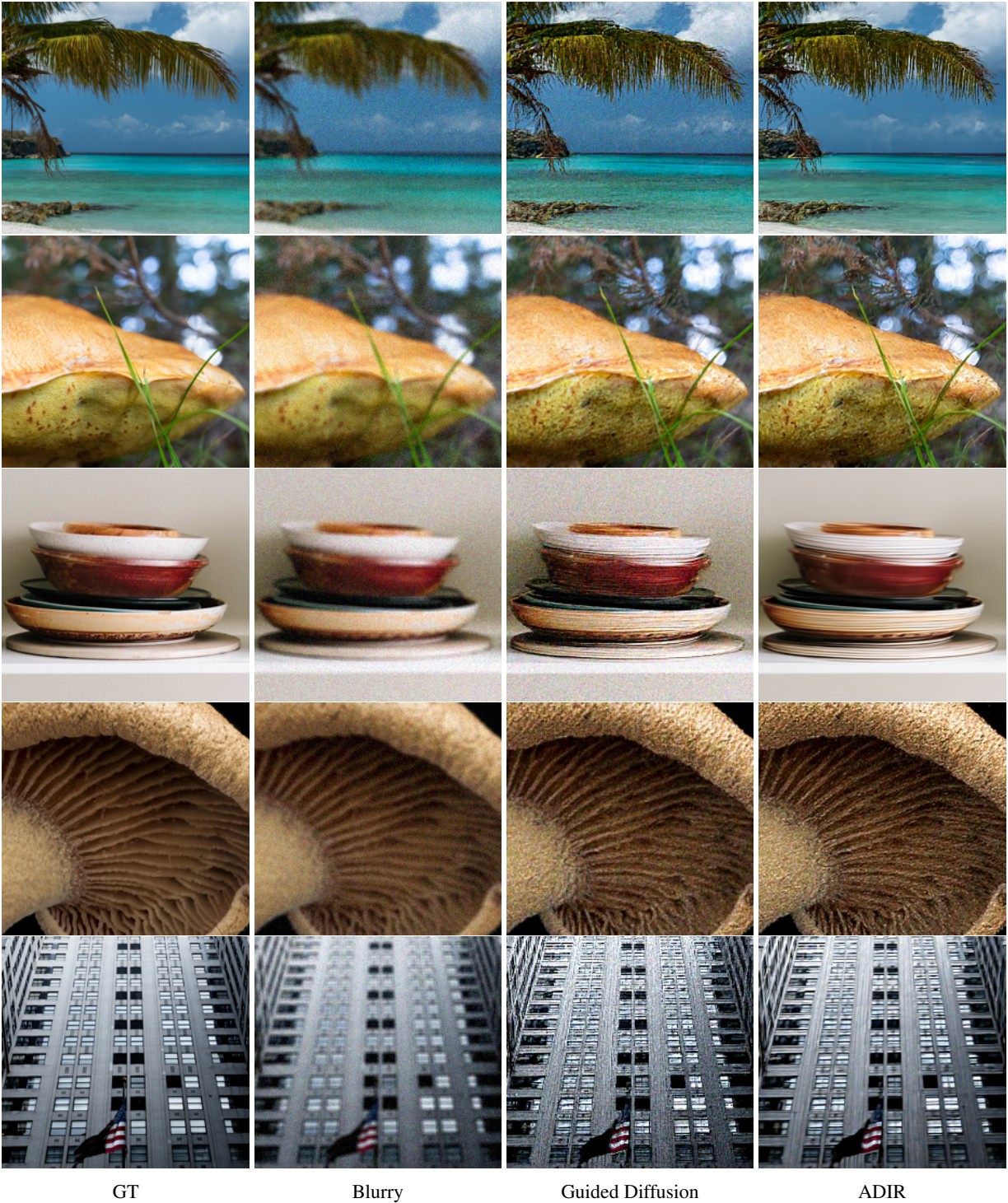

GT       Blurry     Guided Diffusion    ADIR

Figure 13: Deblurring ($5 \times 5$ box filter, $\sigma = 10$) results of Guided Diffusion from section 3.2 and our method (ADIR), using the unconditional model from (Rombach et al., 2022). As can be seen from the images, our method outperforms guided diffusion in both sharpness and reconstruction details.

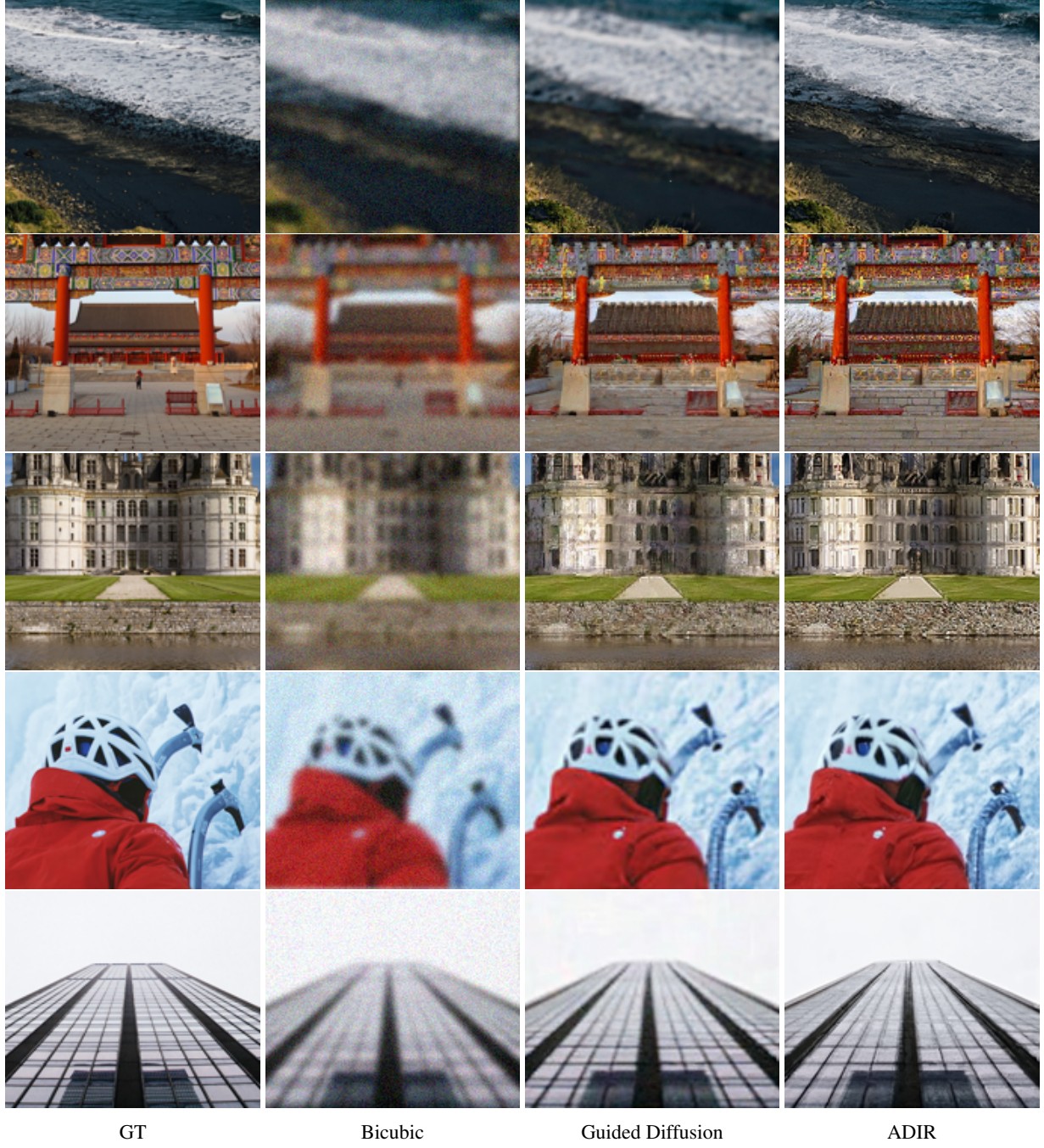

GT       Bicubic     Guided Diffusion     ADIR

Figure 14: Gaussian deblurring ($\sigma_{\text{blur}} = 2$ and $\sigma_{\text{noise}} = 10$) results of Guided Diffusion from section 3.2 and our method (ADIR), using the unconditional model from (Rombach et al., 2022). As can be seen from the images, our method outperforms guided diffusion in both sharpness and reconstruction details.

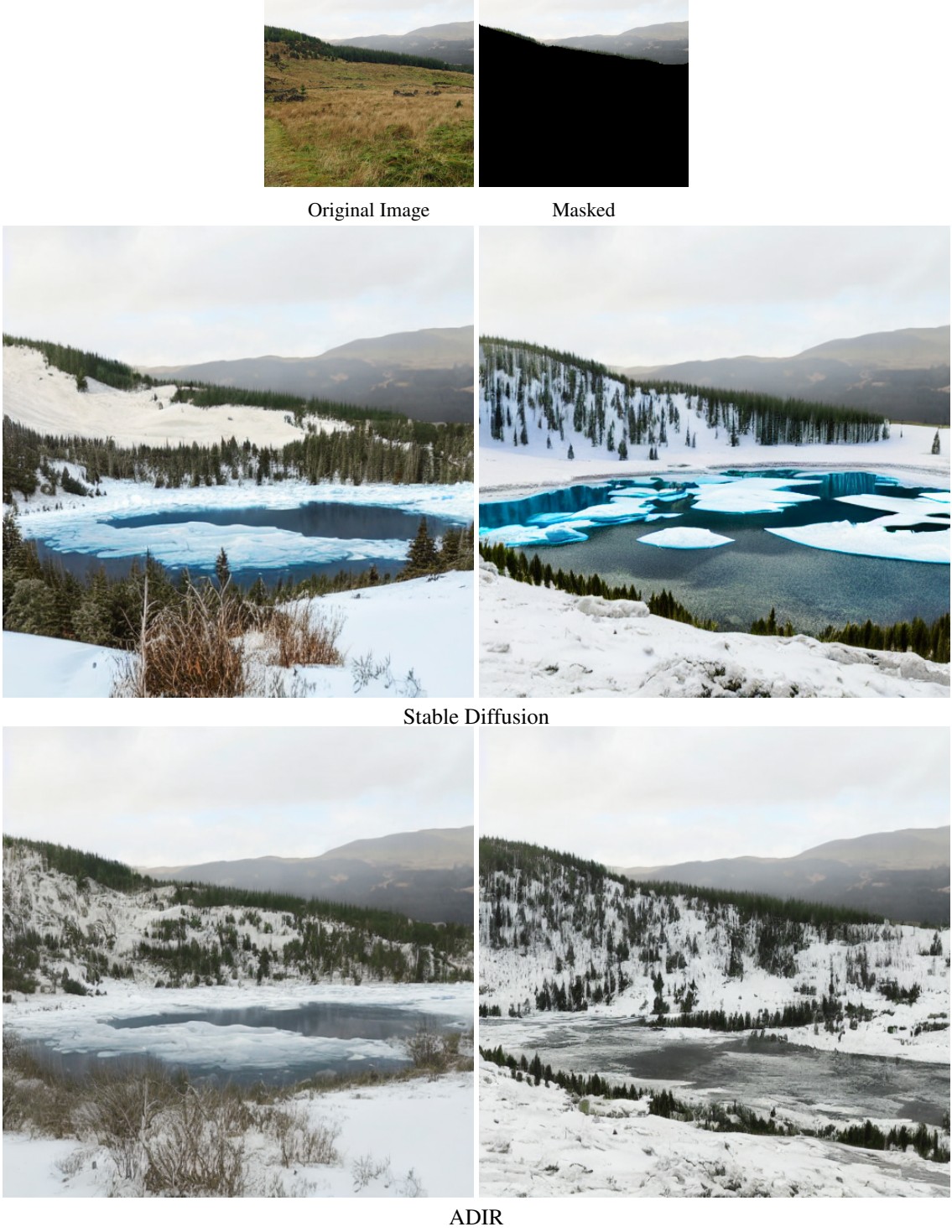

Original Image                    Masked

Stable Diffusion

ADIR

Figure 15: Text-based image editing comparison between Stable Diffusion (Rombach et al., 2022) and ADIR, using the prompt "A beautiful frozen lake between mountains in the snow" for two different seeds.

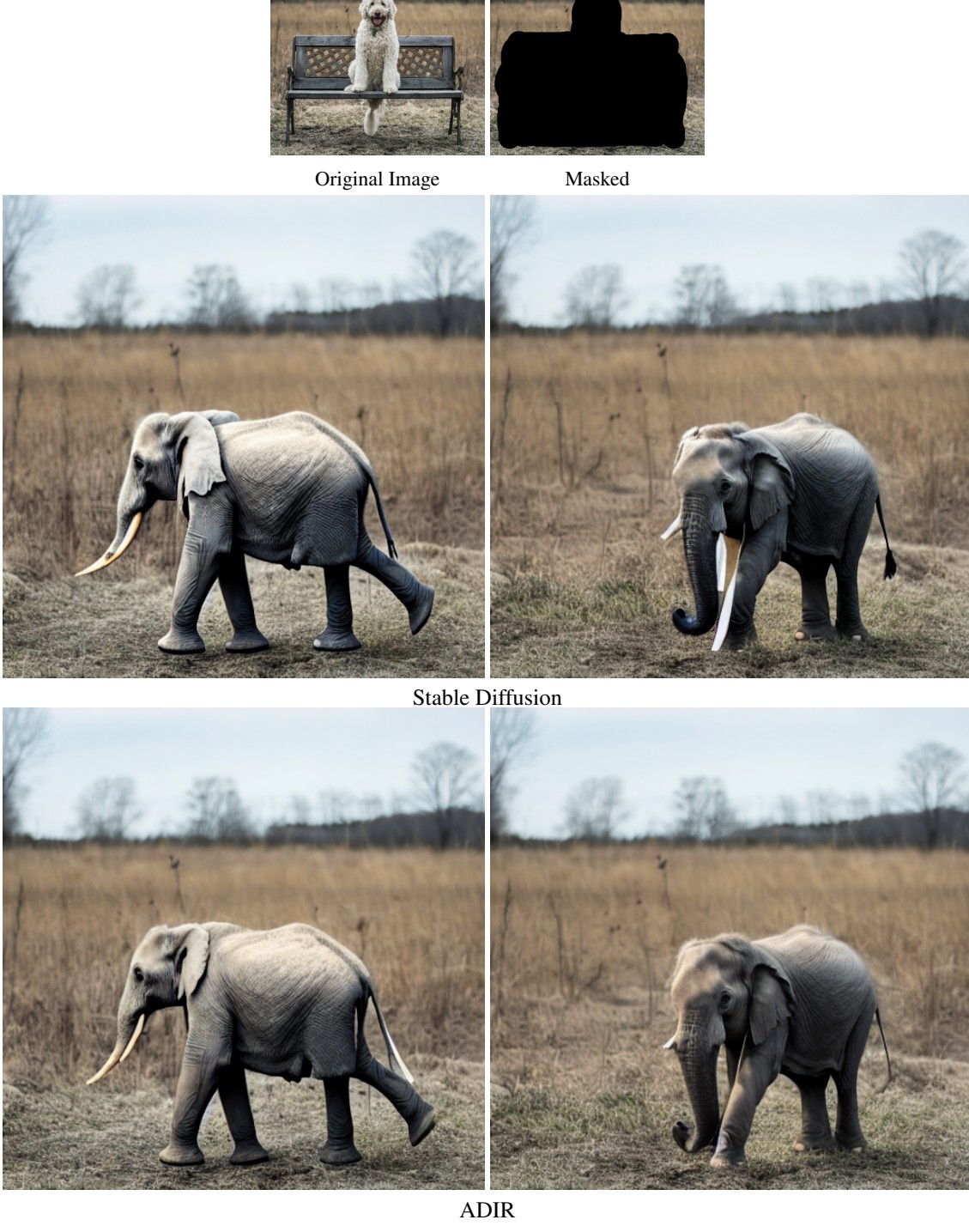

Figure 16: Text-based image editing comparison between Stable Diffusion (Rombach et al., 2022) and ADIR, using the prompt "An elephant walking" for two different seeds.

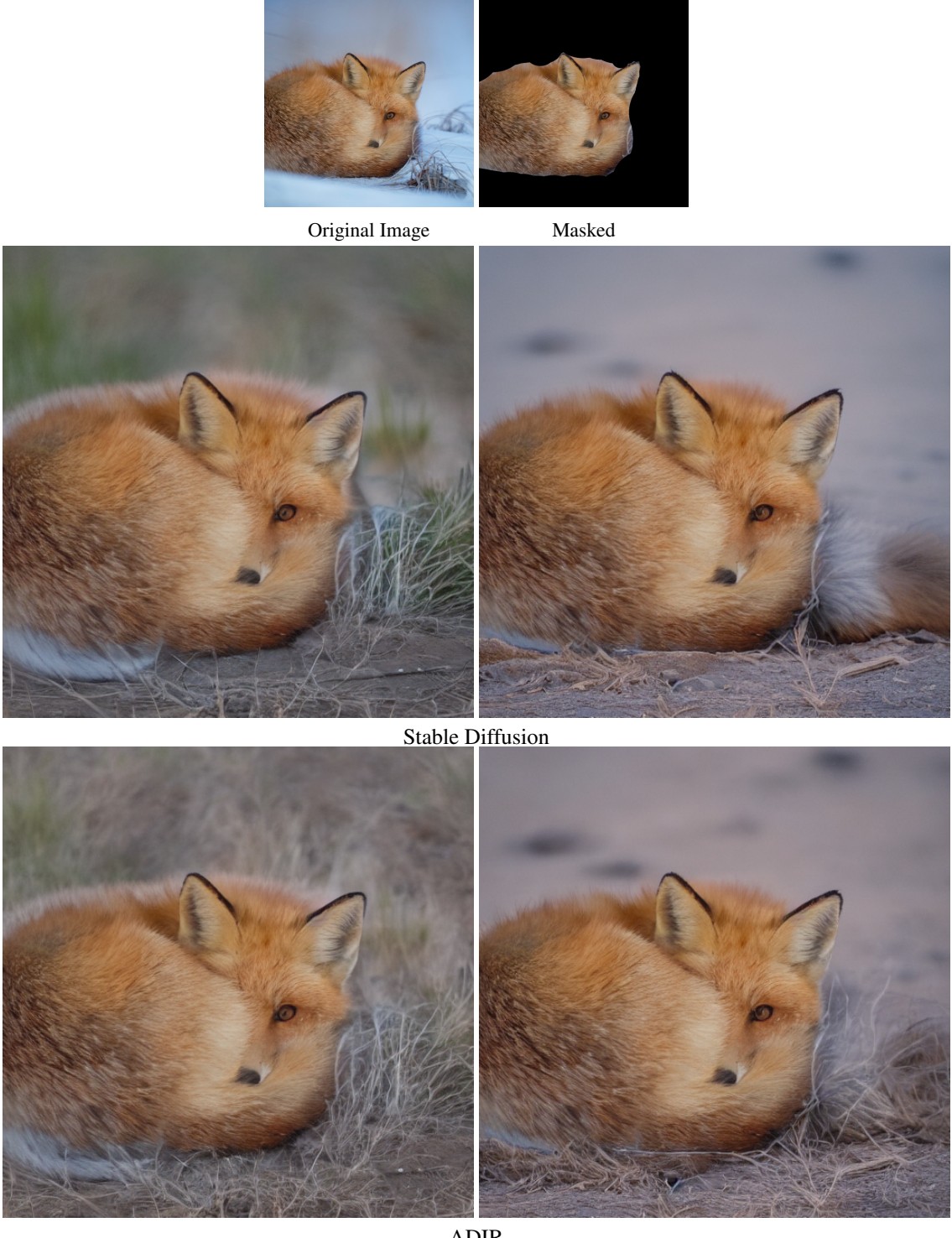

Original Image          Masked

Stable Diffusion

ADIR

Figure 17: Text-based image editing comparison between Stable Diffusion (Rombach et al., 2022) and ADIR applied to the Stable Diffusion model, for the prompt "A fox sitting in the middle of the desert"

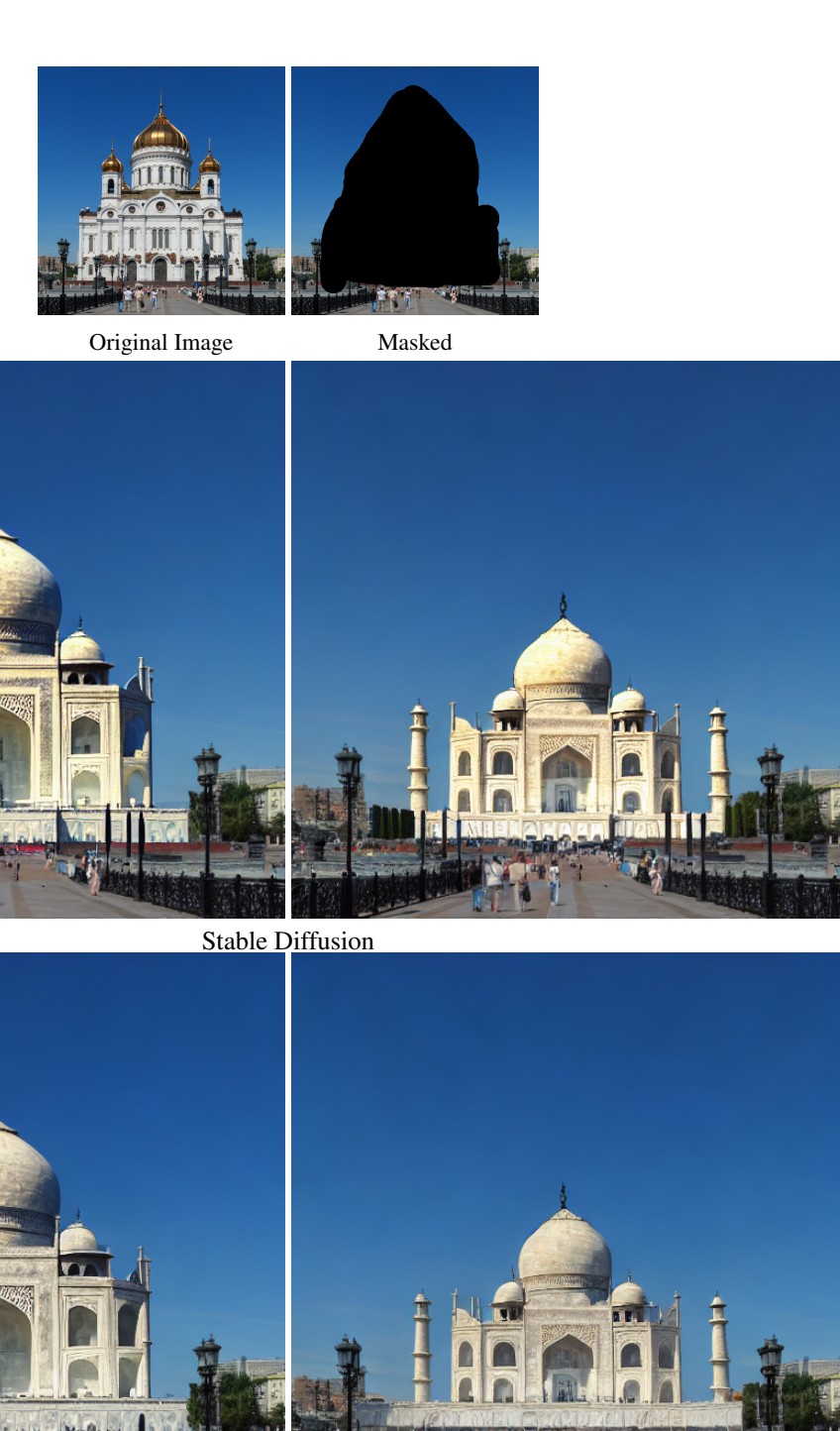

Original Image       Masked

Stable Diffusion

ADIR

Figure 18: Text-based image editing comparison between Stable Diffusion (Rombach et al., 2022) and ADIR applied to the Stable Diffusion model, for the prompt "Taj Mahal"

"A vase of flowers on the table of a living room"

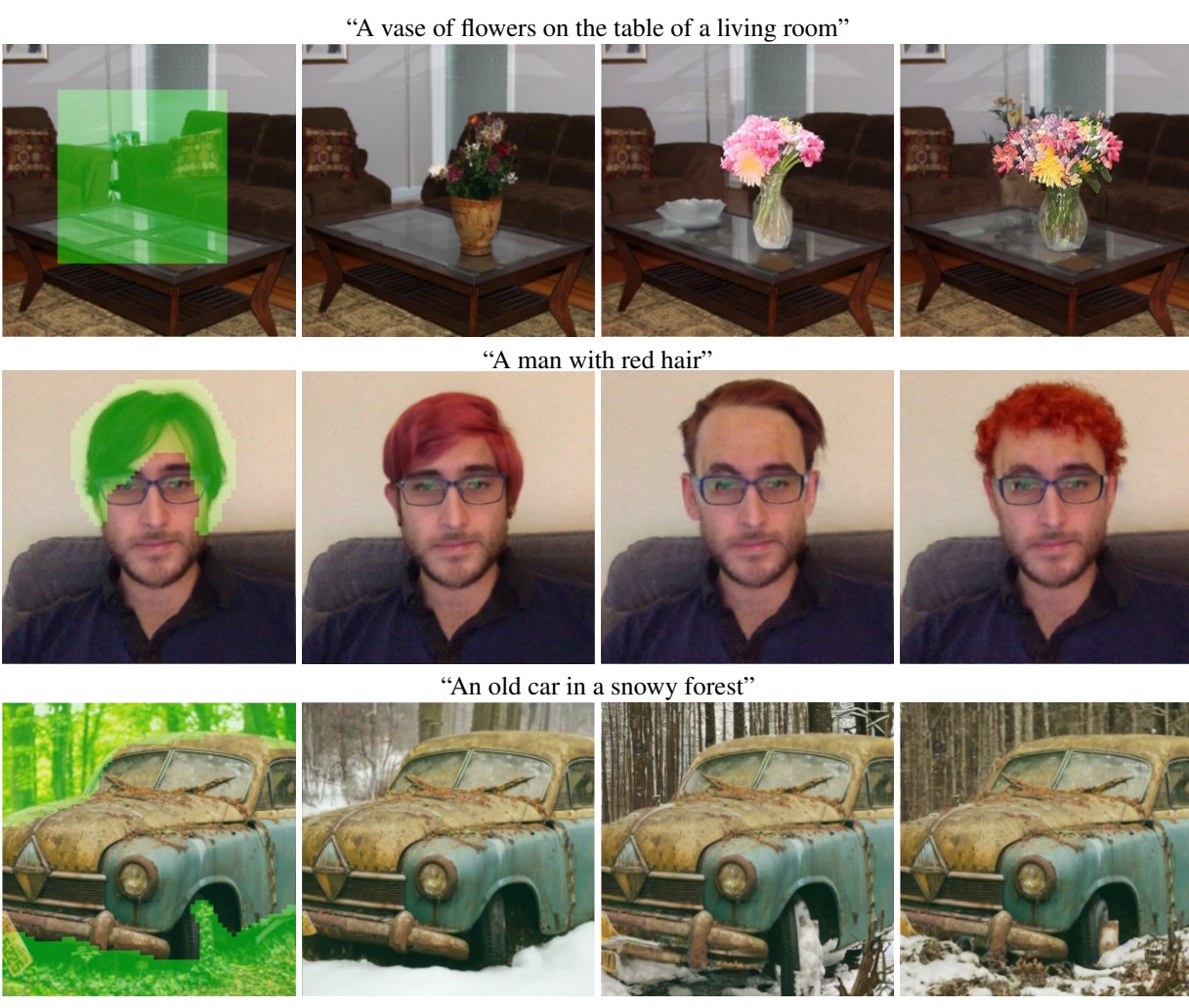

Figure 19: Text-based image editing comparison between GLIDE (full) (Nichol et al., 2021), Stable Diffusion (Rombach et al., 2022) and ADIR applied to the Stable Diffusion model. The images are taken from (Nichol et al., 2021), since their official high-res model was not publicly released. As can be seen, our method produces more realistic images in cases where Stable Diffusion either was not accurate (brown hair instead of red) or in terms of artifacts.

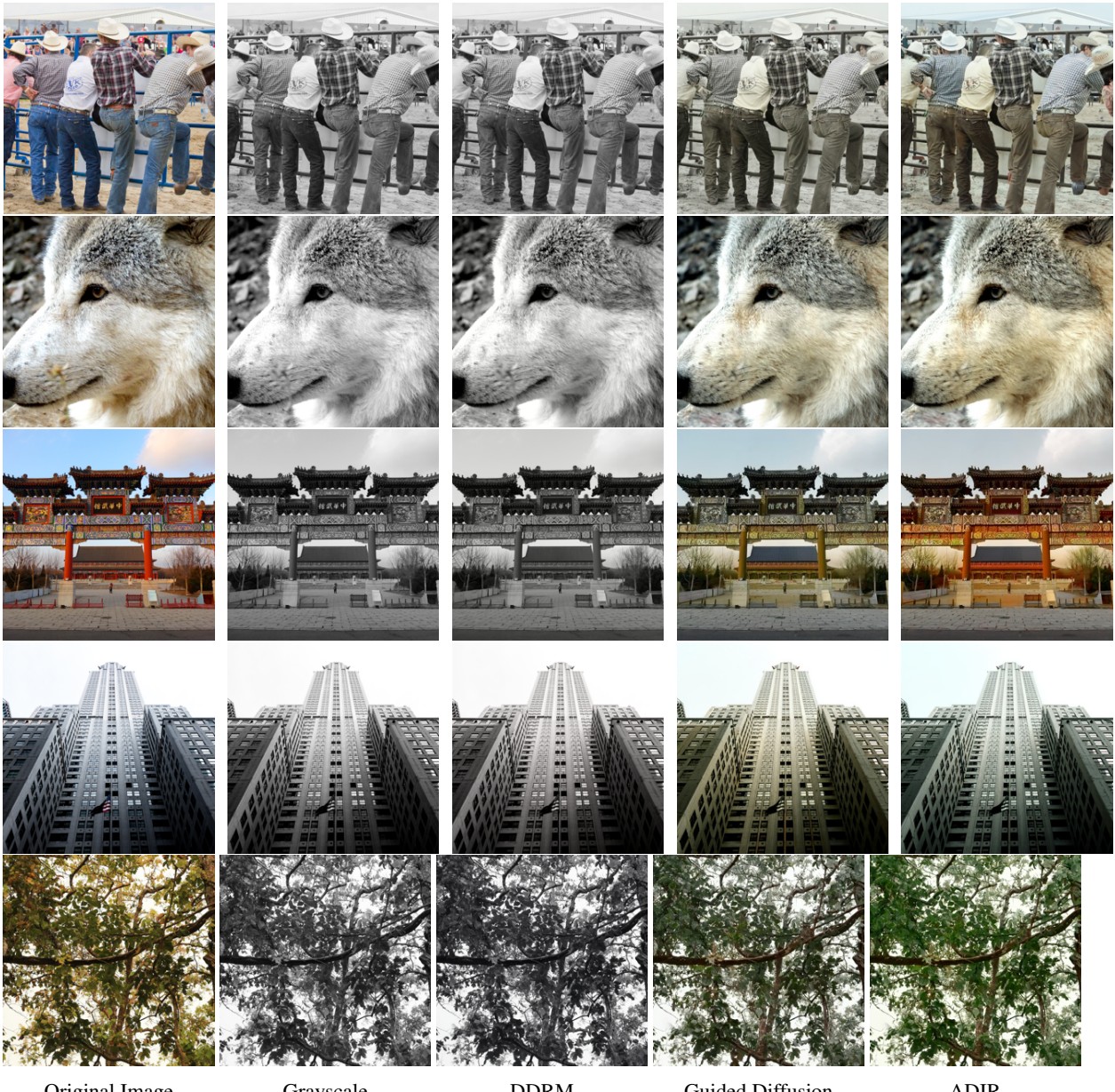

   Original Image        Grayscale        DDRM        Guided Diffusion        ADIR

Figure 20: Image colorization results comparison between DDRM (Kawar et al., 2022a), Guided diffusion proposed in section 3.2, and our adaptive approach ADIR. As can be seen, adapting the denoiser network to the given image can improve the results significantly.

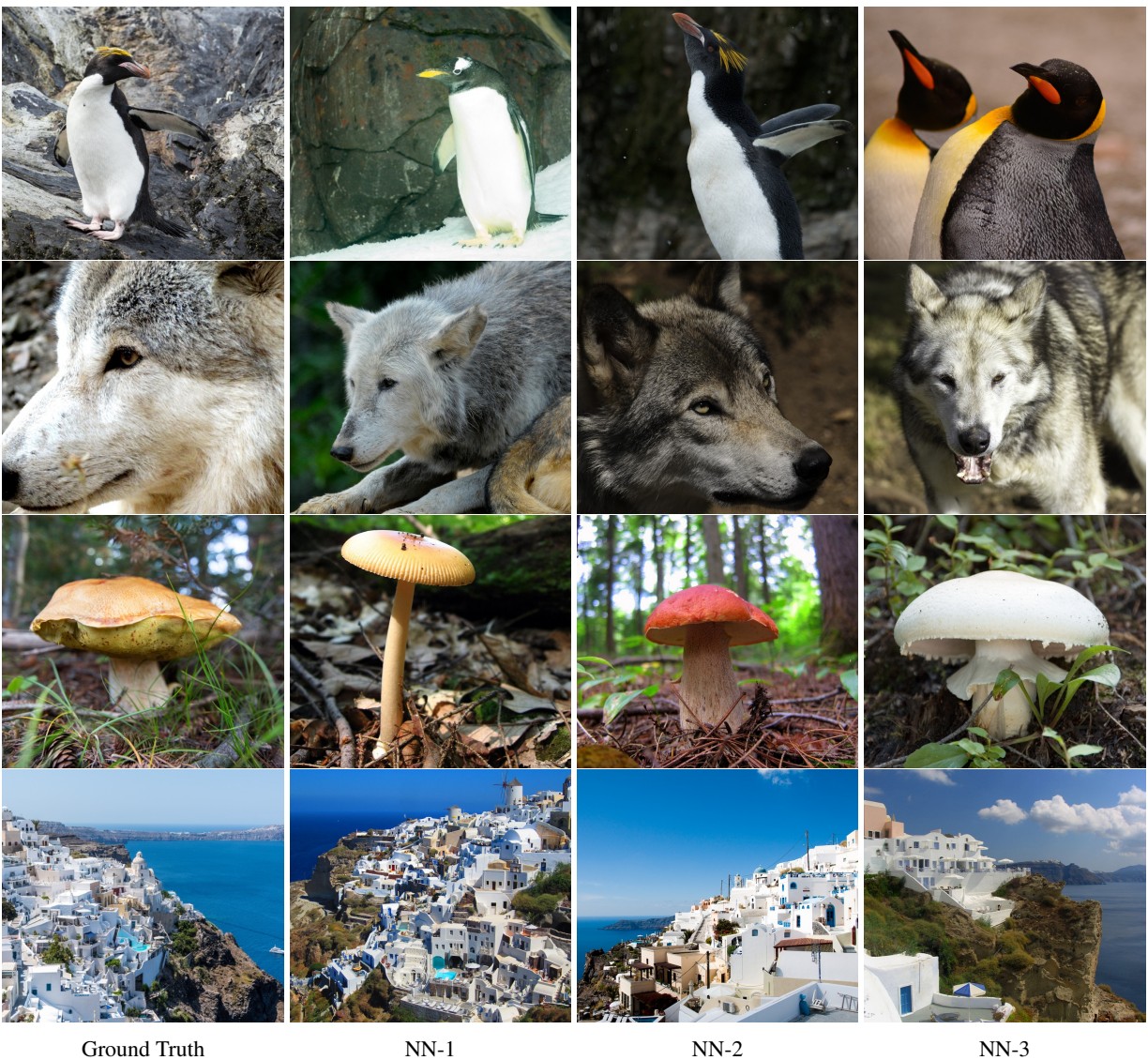

Ground Truth          NN-1          NN-2          NN-3

Figure 21: Examples of images retrieved from Google Open Dataset (Kuznetsova et al., 2020) using CLIP (Radford et al., 2021) for super resolution with scale factor of 8 ($64^2 \rightarrow 512^2$).

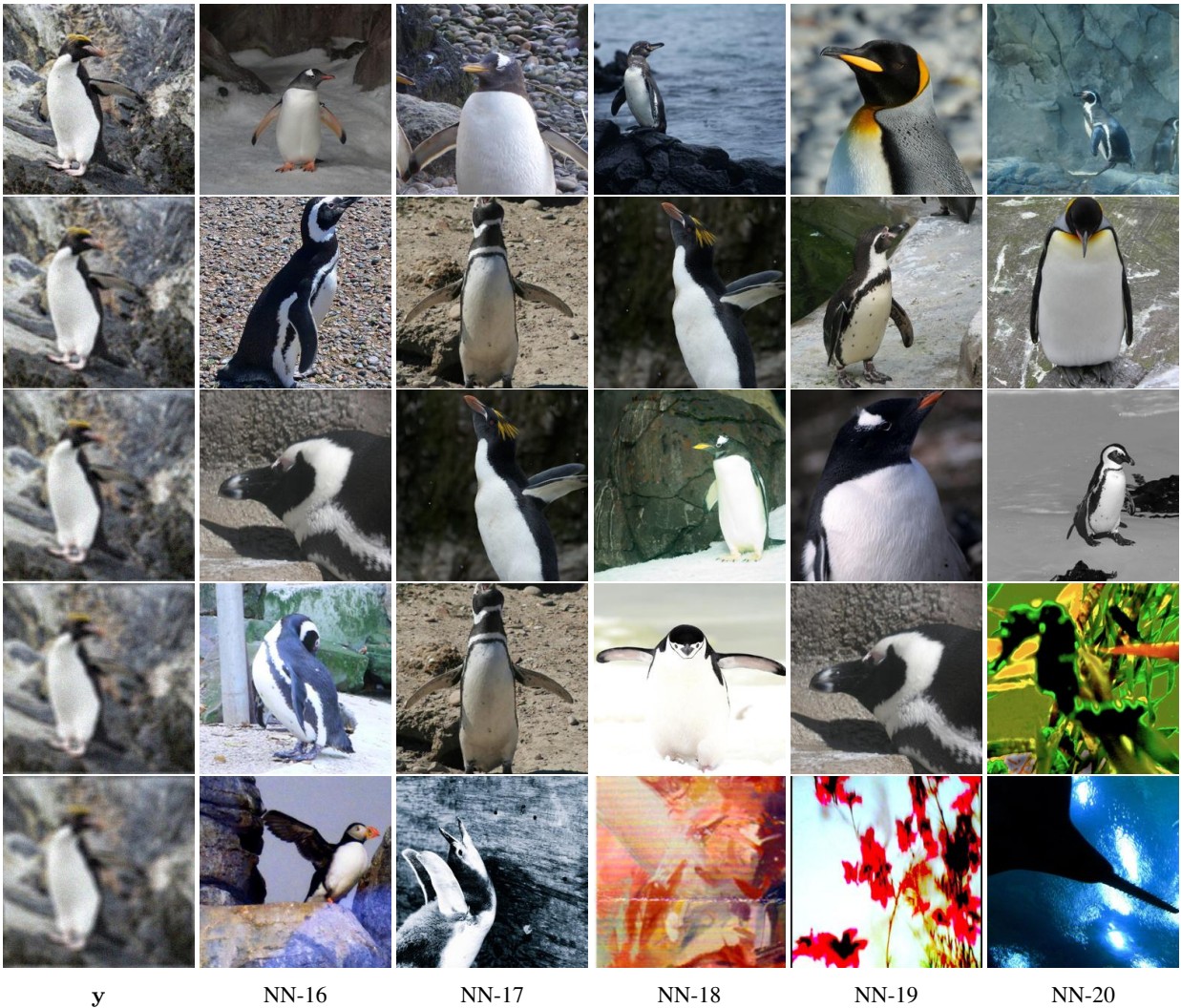

Figure 22: The effect of **A** on the K-NN retrieval: Each row represent a different blur operator **A**, and each column shows the 5 least similar images from the 20 retrieved nearest images from Google Open Dataset (Kuznetsova et al., 2020) using CLIP (Radford et al., 2021). In all cases we used a box filter with support $3 \times 3, 5 \times 5, 7 \times 7, 9 \times 9$ and $11 \times 11$, respective to the row number.

