# OpenReview forum: "ADIR: Adaptive Diffusion for Image Reconstruction"
_TMLR — Rejected by TMLR_

### Review · Reviewer_ZuNZ · 2024-03-20

**Summary Of Contributions:**

This paper presents an algorithm to use pretrained denoising-diffusion models for image restoration. Authors introduce an approach to adapt guided diffusion towards sampling images also with fidelity to a small set of images found to be similar to the image to be restored (via K-NN in a CLIP embedding space). Experiments are conducted for image super-resolution, deblurring, colorization and text-guided image editing.

**Audience:**

Yes

**Claims And Evidence:**

Yes

**Requested Changes:**

- Visual comparisons should be improved (mainly in image deblurring and super-resolution), such that one has zoomed-in boxes of the same region in compared images. Then it can show detailed differences, since sometimes images look pretty much the same with Stable Diffusion and ADIR.

- Description of DDPMs in Section 3.1 states that the noise schedule ends with \beta_T=1. I believe this is an implementation tweak for a given choice of the schedule (e.g., cosine), and does not really have any theoretical reasoning as to why it should always be the case (e.g., in the linear schedule it is often not the case). Perhaps this should be clarified/discussed, perhaps accordingly with this work:
"Common Diffusion Noise Schedules and Sample Steps are Flawed", WACV2024.

- In their Related Work section authors talk about the closest works to theirs, however many diffusion modeling based recent studies that have also explored image restoration are not discussed (especially ones that use pretrained models too):

[a] "Denoising diffusion models for plug-and-play image restoration.", CVPRW 2023.

[b] "Restoring vision in adverse weather conditions with patch-based denoising diffusion models", TPAMI 2023.

[c] "Inversion by direct iteration: an alternative to denoising diffusion for image restoration", TMLR 2023.

**Strengths And Weaknesses:**

Strengths: Authors' approach in adapting pre-trained diffusion models for image reconstruction using selected K-NN images appears to be novel. The paper is technically sound and supports its claims.

Weaknesses: The paper appears to be substantially improved from its previous review phase. However there are only minor issues remaining regarding the presentation of results and related work descriptions.

---

### Review · Reviewer_2BiQ · 2024-04-13

**Summary Of Contributions:**

This paper proposes to use off-the-shelf diffusion models for image reconstruction tasks, such as super-resolution, deblurring, colourization, and editing. The method proposes to use the low-resolution image as a reference to guide the sampling. A kNN retrieval is performed to gather images similar to the test image and fine-tune the diffusion model for better adaptive reconstruction.

**Audience:**

Yes

**Claims And Evidence:**

No

**Requested Changes:**

Please provide the computational cost comparison for the inference stage.

**Strengths And Weaknesses:**

Strengths:
1. The paper is well-written in general. The visualization is illustrative.
2. The experimental section is relatively convincing with comparisons to other methods and ablation studies.

Weaknesses:
1. Both the kNN retrieval and the adaptive fine-tuning are conducted at the inference stage, which is very computationally expensive. Please provide a comparison with other methods regarding the running time per test image including the the kNN retrieval and the adaptive fine-tuning.
2. The proposed guidance method in Algorithms 1 is similar to some other baseline methods, such as "ILVR: Conditioning Method for Denoising Diffusion Probabilistic Models".

---

### Review · Reviewer_fdJV · 2024-04-29

**Summary Of Contributions:**

In this paper, the authors propose a conditional sampling scheme that exploits the prior learned by diffusion models while retaining agreement with the measurements. The proposed scheme performs adaptation using images that are nearest neighbors to the degraded image, retrieved from a diverse dataset using an off-the-shelf visual-language model. The proposed scheme is tested on  Stable Diffusion and Guided Diffusion. Extensive experiments have been performed to verify the effectiveness of the proposed scheme.

**Audience:**

Yes

**Broader Impact Concerns:**

There is no broader impact section. Adding one might be better because the proposed text-guided image editing technique can be used to alter real images with negative societal consequences.

**Claims And Evidence:**

Yes

**Requested Changes:**

1. In Equation 17, the gradient is not correct. It’d be great if the authors could provide the exact expression and then write the approximation. The gradient of Eq. (16) is taken wrt $x_t $, but $y_t$ is also a function of $x_t$ because $y_t$ implicitly depends on $\epsilon_\theta(x_t,t)$.

2. Finetuning a pre-trained generative model (sometimes with billions of parameters) for each measurement and each task is computationally expensive. This is not a practically viable solution. The proposed LoRA finetuning might circumvent the issue of trainbale parameter count, but the requirement for each measurement and each task is demanding.

3. The authors implicitly assume that there exists an off-the-shelf dataset containing nearest neighbors of the given measurements. This is a very strong assumption. What if the proposed retriever returns none? Is there any pre-defined threshold to select NN-images? Or always a fixed number of images are picked from the top of NN-images? Always picking the a fixed set may not always be relevant for the downstream tasks. How would the authors overcome this critical issue?

**Strengths And Weaknesses:**

### Strengths

1. The paper is well-written overall and all the details are presented nicely.

2. Extensive experimental results have been provided on faces, non-faces, outdoor scenes, and text-guided image editing.

3. Quantitative results seem reasonable and sometimes better than prior works, e.g. SRx4 in Table 3.

### Weaknesses

1. In Equation 17, the gradient is not correct. It’d be great if the authors could provide the exact expression and then write the approximation. The gradient of Eq. (16) is taken wrt $x_t $, but $y_t$ is also a function of $x_t$ because $y_t$ implicitly depends on $\epsilon_\theta(x_t,t)$.

2. Finetuning a pre-trained generative model (sometimes with billions of parameters) for each measurement and each task is computationally expensive. This is not a practically viable solution. The proposed LoRA finetuning might circumvent the issue of trainbale parameter count, but the requirement for each measurement and each task is demanding.

3. The authors implicitly assume that there exists an off-the-shelf dataset containing nearest neighbors of the given measurements. This is a very strong assumption. What if the proposed retriever returns none? Is there any pre-defined threshold to select NN-images? Or always a fixed number of images are picked from the top of NN-images? Always picking the a fixed set may not always be relevant for the downstream tasks. How would the authors overcome this critical issue?

---

> ### Author Response · Authors · 2024-05-06
>
> We are grateful for the reviewer's insightful comments. Our response is provided below.
>
> 1.	Please note that in the original version we state above Equation 17 that: “We further assume that $\epsilon_{\theta}$ is independent of $x_t$, which we found to be sufficient in our use-cases.” Based on this, the derivative of $y_t$ with respect to $x_t$ vanishes and we get Equation 17, which we named the “surrogate” gradient.Following the reviewer’s comment, in order to make it clearer, in the revision we include the following full gradient expression:
> $$g = -(2A^T(A x_t - y_t) - 2 \cancel{\frac{\partial}{\partial x_t} y_t^T}(A x_t - y_t),$$
> where we cancel the second term (which includes $\frac{\partial}{\partial x_t} y_t^T$) and thus establish our surrogate gradient, which appeared in Equation 17 of the original paper. Note that the reason we make this simplifying assumption is that our empirical investigation showed that the computationally demanding term that we omit, which requires back-propagating through the model and essentially computing its Jacobian, does not lead to any performance benefit. It is interesting to note that other recent works also promote avoiding the computation of the model’s Jacobian. For example, see how the loss is simplified in [1] and the comparison with DPS (which computes the Jacobian) in [2].
>
> 2.	Indeed, as we show in Table 6 of the revised version and mention in the limitation section, the adaptation phase requires additional computations in addition to the diffusion sampling part; yet, we think that it is compensated by the reconstruction performance achieved when adapting the denoiser network. In the revision, we state the runtime optimization as a direction for future research. Note that reducing the test-time tuning runtime is an ongoing problem that is relevant also beyond the diffusion model literature [3], and advances there are complementary to the contribution of our paper.
>
> 3.	Our retrieval approach assumes that the dataset is diverse and that it includes the test images domain. In our test scenarios, we did not need a predefined threshold since the dataset we used contains 1.7M generic scene images. This size was sufficient for retrieving similar images. Yet, we acknowledge that there exist other specific data domains where there might NN retrieval might be difficult. In the limitation section, we added a paragraph discussing this issue that suggests handling this question as a subject for future research.
>
> [1] DreamFusion: Text-to-3D using 2D Diffusion; Ben Poole, Ajay Jain, Jonathan T. Barron, and Ben Mildenhall, ICLR 2023.
> [2] Image Restoration by Denoising Diffusion Models with Iteratively Preconditioned Guidance; Tomer Garber, and Tom Tirer, CVPR 2024.
> [3] Tirer, Tom, Raja Giryes, Se Young Chun, and Yonina C. Eldar. "Deep Internal Learning: Deep Learning from a Single Input." IEEE Signal Processing Magzine, 2024.

---

### Decision · Action_Editor_7fmP · 2024-07-03

**Recommendation:** Reject

**Comment:**

Three expert reviewers reviewed this paper.

All the reviewers acknowledged the technical novelty of adapting pre-trained diffusion models for image restoration tasks using K-NN-retrieved images. They also think the presented experiments show improved results over the compared method via quantitative evaluation.

The reviewers' main concerns include
a) high computational complexity (all reviewers),
- in the revision, the authors provided Table 6. However, as pointed out by Reviewer 2BiQ, the results include only the baselines and not other state-of-the-art methods. In particular, the time cost of the proposed method is extremely demanding (1308 seconds/image). It's unclear why the baseline Guided Diffusion also has a high cost (803 seconds/image).

b) the assumption of a relevant dataset for retrieval (Reviewer fdJV)
- the authors clarify this point by providing more details about using a dataset of 1.7M generic scene images.

c) missing citations/discussions/comparisons of prior diffusion-based restoration methods (Reviewer 2BiQ, Reviewer ZuNZ)
- the authors added citations for the mentioned papers, but did not provide comparisons with these state-of-the-art. These methods do not require computationally expensive test-time fine-tuning. Performance improvement over these methods needs to be clearly demonstrated.

After reading the authors' responses, the final recommendations are
- Reviewer 2BiQ: Leaning reject
- Reviewer ZuNZ: Leaning reject
- Reviewer fdJV: Leaning accept

all three reviewers "Weakly Oppose" to recommend to ICLR journal-to-conference track.

Considering the strengths and weaknesses outlined by the reviewers, the AE agrees with Reviewer 2BiQ and ZuNZ that the time complexity comparisons and the restoration performance comparisons with highly relevant methods should be included for full contexts. The paper, in its current state, is not yet ready for TMLR publication.

**Audience:**

Yes, the paper is relevant to the TMLR's audience.

**Claims And Evidence:**

This paper proposes a conditional sampling scheme for image restoration tasks (e.g., super-resolution, colorization). This is achieved by fine-tuning (with LoRA) a pre-trained diffusion model using images retrieved from a diverse dataset with the degraded image as a query. The paper applied the proposed adaptive diffusion to two base models, stable diffusion and guided diffusion, and reported deblurring, super-resolution, and colorization results.

From the reviewers' comments, there are three specific aspects where the evidence is not sufficiently strong.

1) Time-complexity.
- In the revised Table 6, the authors provide the time-cost comparison, but only with the baselines. No other state-of-the-art diffusion-based restoration methods are compared. These methods do not require computationally intensive fine-tuning and thus run much faster.

2) Missing comparisons with state-of-the-art
- Reviewer ZuNZ provided three references of more recent diffusion-based restoration models that do not require test-time fine-tuning. The authors added citations in the revised version but did not provide a comparison (in both visual results and runtime comparison). Comparison with these methods would be important to support the claim regarding the benefits of the proposed test-time fine-tuning scheme.

2) Visual comparisons with the baselines.
- In the revised paper, the authors provided zoom-in patches in Figure 5, Figure 6 and several other figures in the appendix. While there are subtle improvements quantitatively, the visual results are virtually the same as the baseline (even at the zoom-in levels). The benefits of costly test-time adaptation are thus unconvincing.

**Resubmission Of Major Revision:**

The authors may consider submitting a major revision at a later time.